# Nucleolar release of rDNA repeats for repair involves SUMO-mediated untethering by the Cdc48/p97 segregase

Matías Capella [1,2,7 ✉], Imke K. Mandemaker [1], Lucía Martín Caballero[1,3], Fabian den Brave [2,6], Boris Pfander [3,4], Andreas G. Ladurner[1,3], Stefan Jentsch[2] & Sigurd Braun [1,3,5,7 ✉]

Ribosomal RNA genes (rDNA) are highly unstable and susceptible to rearrangement due to their repetitive nature and active transcriptional status. Sequestration of rDNA in the nucleolus suppresses uncontrolled recombination. However, broken repeats must be first released to the nucleoplasm to allow repair by homologous recombination. Nucleolar release of broken rDNA repeats is conserved from yeast to humans, but the underlying molecular mechanisms are currently unknown. Here we show that DNA damage induces phosphorylation of the CLIP-cohibin complex, releasing membrane-tethered rDNA from the nucleolus in *Saccharomyces cerevisiae*. Downstream of phosphorylation, SUMOylation of CLIP-cohibin is recognized by Ufd1 via its SUMO-interacting motif, which targets the complex for disassembly through the Cdc48/p97 chaperone. Consistent with a conserved mechanism, UFD1L depletion in human cells impairs rDNA release. The dynamic and regulated assembly and disassembly of the rDNA-tethering complex is therefore a key determinant of nucleolar rDNA release and genome integrity.

[1] Department of Physiological Chemistry, BioMedical Center (BMC), Ludwig-Maximillians-Universität München, Planegg-Martinsried, Germany. [2] Max Planck Institute of Biochemistry, Molecular Cell Biology, Planegg-Martinsried, Germany. [3] International Max Planck Research School for Molecular and Cellular Life Sciences, Planegg-Martinsried, Germany. [4] Max Planck Institute of Biochemistry, DNA Replication and Genome Integrity, Planegg-Martinsried, Germany. [5] Institute for Genetics, Justus-Liebig University Giessen, Giessen, Germany. [6] Present address: Institute for Biochemistry and Molecular Biology, University of Bonn, Bonn, Germany. [7] These authors jointly supervised this work: Matías Capella, Sigurd Braun. ✉email: matias.capella@bmc.med.lmu.de; sigurd.braun@gen.bio.uni-giessen.de

Eukaryotic genomes contain large amounts of repetitive sequences at centromeres, telomeres, and ribosomal RNA genes (rDNA). In *Saccharomyces cerevisiae*, the rDNA locus consists of ~150 copies, organized in 9.1 kb tandem repeats on chromosome XII. Humans have hundreds of 43 kb rDNA units located on different chromosomes. This abundance of donor sequences means that rDNA repeats often undergo homologous recombination (HR), making them a major source of genome instability[1]. As an example, DNA double-strand break (DSB) repair within a repetitive array can cause repeat insertions or deletions through HR with a neighboring unit[1,2]. Such rearrangements can have severe consequences. Indeed, rDNA repeat translocations are associated with cellular senescence, neurodegenerative diseases, and are amongst the most common events observed in cancer cells[3–5].

rDNA repeats are spatially segregated into the nucleolus, a membrane-less subnuclear organelle. In *S. cerevisiae*, rDNA is tethered to the nuclear envelope by the CLIP (chromatin linkage of inner nuclear membrane proteins) and the cohibin complexes[6]. Cohibin is a "V"-shaped complex composed of one Lrs4 homodimer and two Csm1 homodimers. It associates with rDNA through its interaction with the RENT complex (regulator of nucleolar silencing and telophase exit) and Tof2[6–8]. Cohibin tethers rDNA to the nuclear periphery by binding the CLIP complex. CLIP comprises two integral inner nuclear membrane proteins, the LEM domain protein Heh1 (also known as Src1) and Nur1 (Supplementary Fig. 1a)[6]. Loss of CLIP or cohibin promotes rDNA release from the nucleolus into the nucleoplasm, resulting in increased recombination among rDNA repeats[6].

The small ubiquitin-like modifier SUMO plays multiple roles in DNA damage repair[9]. Akin to ubiquitylation, an enzymatic cascade covalently attaches SUMO to specific lysine residues on target proteins. SUMO and its conjugates are recognized by proteins containing SUMO interacting motifs (SIMs)[10]. Furthermore, several SUMO moieties can be conjugated to form poly-SUMO chains, which trigger protein degradation through the recruitment of STUbLs (SUMO-targeted ubiquitin ligases)[11]. Conversely, SUMO moieties can be removed by the SUMO-specific proteases Ulp1 and Ulp2, which also mediate SUMO maturation by exposing the C-terminal Gly-Gly motif[10].

DNA repair often requires the movement of the damaged site to another compartment. Several types of DNA damage, including persistent DSBs, collapsed replication forks, eroded telomeres, and breaks within heterochromatin, are relocated to the nuclear periphery in a SUMOylation-dependent manner to promote alternative repair pathways[12–17]. The yeast STUbl Slx5/Slx8 and its homologs in flies and mammals recruit poly-SUMOylated DNA damage sites to nucleopores independently of the recombinase Rad51[12,14,15,17,18]. A different relocation pathway employs mono-SUMOylation and Rad51-dependent targeting of repair intermediates to the SUN (Sad1-Unc84-related) nuclear membrane protein Mps3[12,13,19], which may help to prevent promiscuous recombination. In addition, rDNA repeats relocate to the nuclear periphery upon DSB induction, which has been proposed to serve as a backup mechanism for DNA repair[20].

While SUMOylation of various repair factors contributes to the relocation of damaged DNA sites[17], it also supresses aberrant rDNA repeat recombination through SUMOylation of Rad52, a key player in DSB repair[6,21]. Together with the Smc5/6 (structural maintenance of chromosomes) and MRX (Mre11-Rad50-Xrs2) complexes, SUMOylation prevents Rad52 from forming recombination foci in the nucleolus[21]. Formation of nucleolar Rad52 foci is further prevented by condensin-mediated rDNA condensation and RNA polymerase I (Pol I) transcription[22]. Keeping rDNA repeats within the nucleolus prevents aberrant recombination between different units; however, this spatial confinement

presents a threat when cells encounter DNA damage. Damaged rDNA repeats need to be released from the nucleolus, and re-enter once repaired by the HR machinery[21,23,24]. Similarly, repair of centromeric heterochromatin is blocked by Smc5/6-dependent exclusion of Rad51 in *Drosophila* and mouse cells, and its recovery involves the release of the damaged locus outside the heterochromatin domain[25,26]. The molecular events underlying the movement of damaged sites to the nuclear periphery have been intensely studied and include rDNA relocation[12–17,20]. However, the regulatory mechanisms controlling the initial release of broken rDNA repeats from the nucleolus remain largely elusive.

Here, we report the molecular events that regulate nucleolar rDNA release in *S. cerevisiae*. We find that a series of post-translational modifications control the association of key molecular players at the nuclear periphery. We show that blocking rDNA relocation using a constitutively bound CLIP-cohibin complex is detrimental for cell survival, even in the absence of external DNA damage. This result implies that CLIP-cohibin dissociation is dynamically regulated. We find that Nur1 phosphorylation is a critical trigger for disruption of the CLIP-cohibin complex (hereafter referred to as rDNA tethering complex). In addition, our data reveal that SUMOylation of CLIP-cohibin mediates the movement of individual rDNA repeats out of the nucleolus. SUMOylated CLIP-cohibin recruits the Cdc48/p97 segregase, thereby promoting the disassembly of the rDNA tethering complex, a mechanism conserved in human cells. We propose that phosphorylation and SUMOylation of the CLIP-cohibin complex facilitate the transient release of damaged rDNA repeats from the nucleolus to the nucleoplasm to allow DNA repair.

## Results

**Permanent rDNA tethering is detrimental for cell survival.** Deletion of CLIP or cohibin causes rDNA release from the nucleolus, which is reminiscent of rDNA relocation observed during DNA damage repair[6,21]. We thus postulated that rDNA release requires dynamic dissociation of CLIP and cohibin. To test this hypothesis, we generated an inducible artificial tethering system by fusing Heh1 (CLIP) to GBP (GFP-binding protein), which was expressed under the control of the *GAL1* promoter together with [GFP]Lrs4 (cohibin) (Fig. 1a). Intriguingly, when induced by galactose, these fusion proteins caused a severe growth defect, demonstrating that failure to dissociate Heh1 from Lrs4 is toxic for cells (Fig. 1b, lane 4). A similar phenotype was observed when co-expressing GFP-fusions of the rDNA-bound protein Net1[8] with Heh1-GBP (Fig. 1a, b, lanes 6 and 8). Expressing Heh1 directly fused to either Lrs4 or Tof2 also produced severe growth defects (Fig. 1c, lanes 4 and 7). In contrast, perinuclear tethering of the unrelated genomic Gal4-DNA-binding domain did not affect growth (Fig. 1c, lane 8). Notably, growth was fully restored when expressing linear Heh1-Lrs4 fusions with mutations preventing either cohibin assembly (*lrs4 Δ1-35*)[7] or interaction with downstream factors (*lrs4 Q325Stop*)[27] (Fig. 1c, lanes 5 and 6, respectively). We verified that all fusion variants displayed similar expression levels, thus excluding altered protein levels underlying the lack of growth defect. The only exception was Heh1-Tof2 expression, which was undetectable despite the substantial phenotype caused by its expression (Supplementary Fig. 1b). Moreover, the observed phenotypes correlated with the expression levels, as driving Heh1-GBP or Heh1-Lrs4 expression with the weaker *GALS* promoter resulted in less pronounced growth defects (Supplementary Fig. 1c–e). To examine the location of individual repeats in cells expressing Heh1-Lrs4 fusion proteins, we used a strain that harbors a

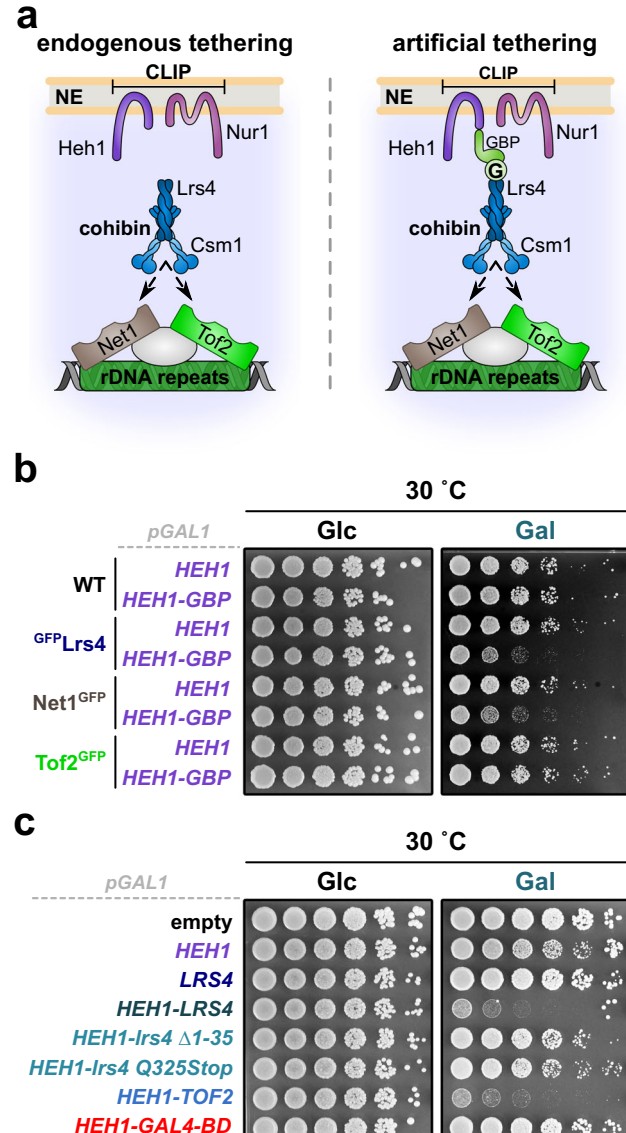

**Fig. 1 Constitutive perinuclear tethering of rDNA is detrimental. a** Scheme of synthetic rDNA tethering through the expression of GFP- and GBP-fusion proteins. Lrs4 binds ribosomal repeats through interaction with the rDNA-bound proteins Net1 or Tof2. G, GFP. **b** Fivefold serial dilutions of WT, GFPLrs4, Net1GFP, and Tof2GFP cells transformed with empty vector or a plasmid bearing *HEH1* fused to GFP-binding protein (*GBP*). **c** Fivefold serial dilutions of WT cells transformed with empty vector or plasmids bearing the indicated GFP-tagged fusion proteins. **b**, **c** Cells were spotted and grown on selective media with 2% glucose (control, Glc) or 2% galactose (induction, Gal) at 30 °C for 3 days. All constructs contain the galactose-inducible *GAL1* promoter.

tandem array of Tet-repressor-binding sites (224x*tetO*) inserted in the rDNA locus, which can be localized using TetI^mRFP expression[21]. This strain also expresses Rad52^YFP and Nop1^ECFP allowing visualization of Rad52 foci and the nucleolus, respectively (Supplementary Fig. 1f). As expected, the expression of Heh1-lrs4 Q325Stop increased the frequency of marked repeats outside the nucleolus (Supplementary Fig. 1f; see "methods" section for details). No difference was detected for cells expressing Heh1 or Heh1-Lrs4, despite the severe growth defect observed for the fusion protein. However, TetI^mRFP localization outside the nucleolus in WT cells occurs only transiently and at a very low

rate (i.e., less than 15%). Thus, a further decrease in cells over-expressing Heh1-Lrs4 might be difficult to detect, also because the non-fused WT proteins are still present. Based on these observations, we conclude that cell viability correlates with the dissociation of ribosomal repeats from the nuclear periphery, and dynamic CLIP-cohibin disassembly.

**The C-terminal domain of Nur1 mediates rDNA tethering complex association**. To unveil the molecular mechanism that controls CLIP-cohibin disassembly, we dissected interactions within the rDNA tethering complex. Using co-immunoprecipitation (coIP) assays, we found that deleting individual CLIP or cohibin components affects the binding of the two complexes. Specifically, Heh1 (CLIP) failed to interact with Csm1 (cohibin) in *lrs4Δ* or *nur1Δ* cells (Fig. 2a, b). Moreover, Lrs4 (cohibin) interaction with Nur1 (CLIP) was abolished in cells lacking Csm1 (Supplementary Fig. 2a). These results indicate that the integrity of both complexes is required for CLIP-cohibin association. In contrast, Heh1 interaction with Nur1 was largely unaffected in the absence of Lrs4 (Supplementary Fig. 2b), revealing that CLIP formation is independent of cohibin integrity. In *S. cerevisiae*, *HEH1* is a rare example of alternative splicing, which generates a shorter version known as Heh1-S that does not bind Nur1[28,29]. We consistently detected Lrs4 binding to Heh1, but not Heh1-S, upon expression of an N-terminal GFP-Heh1 fusion that allows detection of both splice variants (Supplementary Fig. 2c).

Using yeast two-hybrid assays (Y2H), we mapped interactions between individual domains. We found that the N-terminal Heh1 domain is critical for Csm1 association, while the C-terminal Nur1 region interacts with both cohibin members (Fig. 2c). Further analysis narrowed the Nur1 interaction interface with Csm1 down to the last 54 residues of the C-terminus (Fig. 2d, Supplementary Fig. 2d). Using coIP, we confirmed the critical role of these residues for Nur1 association with Lrs4 in vivo (Fig. 2e). Removal of the last residues also abolished Nur1 binding to Sir2 (Supplementary Fig. 2e), which might associate Nur1 through cohibin[8]. In contrast, deletion of these C-terminal Nur1 residues does not impact CLIP complex formation (Supplementary Fig. 2e).

We next monitored whether the truncated Nur1 variants affect the location of individual rDNA repeats (Fig. 2f). Compared to WT cells, mutants lacking the Nur1 C-terminal domain or its last 54 residues displayed a marked increase in the number of cells with TetI^mRFP-marked rDNA foci outside the nucleolus, reaching levels similar to *nur1Δ* cells (Fig. 2g). From these data, we conclude that the most C-terminal region of Nur1 is required for CLIP-cohibin association.

**Phosphorylation of the CLIP complex disrupts rDNA tethering**. Nur1 was shown to be phosphorylated[30], and this modification is removed by the nucleolar phosphatase Cdc14 recognizing the C-terminal domain (Fig. 2d). As the C-terminal domain is also required for cohibin interaction, we tested whether phosphorylation triggers the disassembly of the rDNA tethering complex. Indeed, we found that interaction of Nur1 with Lrs4 was reduced under conditions that favor the accumulation of phosphorylated Nur1, either by using a thermosensitive mutant of Cdc14 (*cdc14-3*) or globally blocking dephosphorylation (Fig. 3a, Supplementary Fig. 3a). Conversely, depleting Net1, which inhibits Cdc14 phosphatase activity[31], increased Nur1 interaction with Lrs4 (Supplementary Fig. 3b). To determine the relevant target sites in Nur1, we mutated several known phosphorylation sites[32] to aspartate thus mimicking the phosphorylated state (Supplementary Fig. 3c). Notably, using Y2H assays, we found

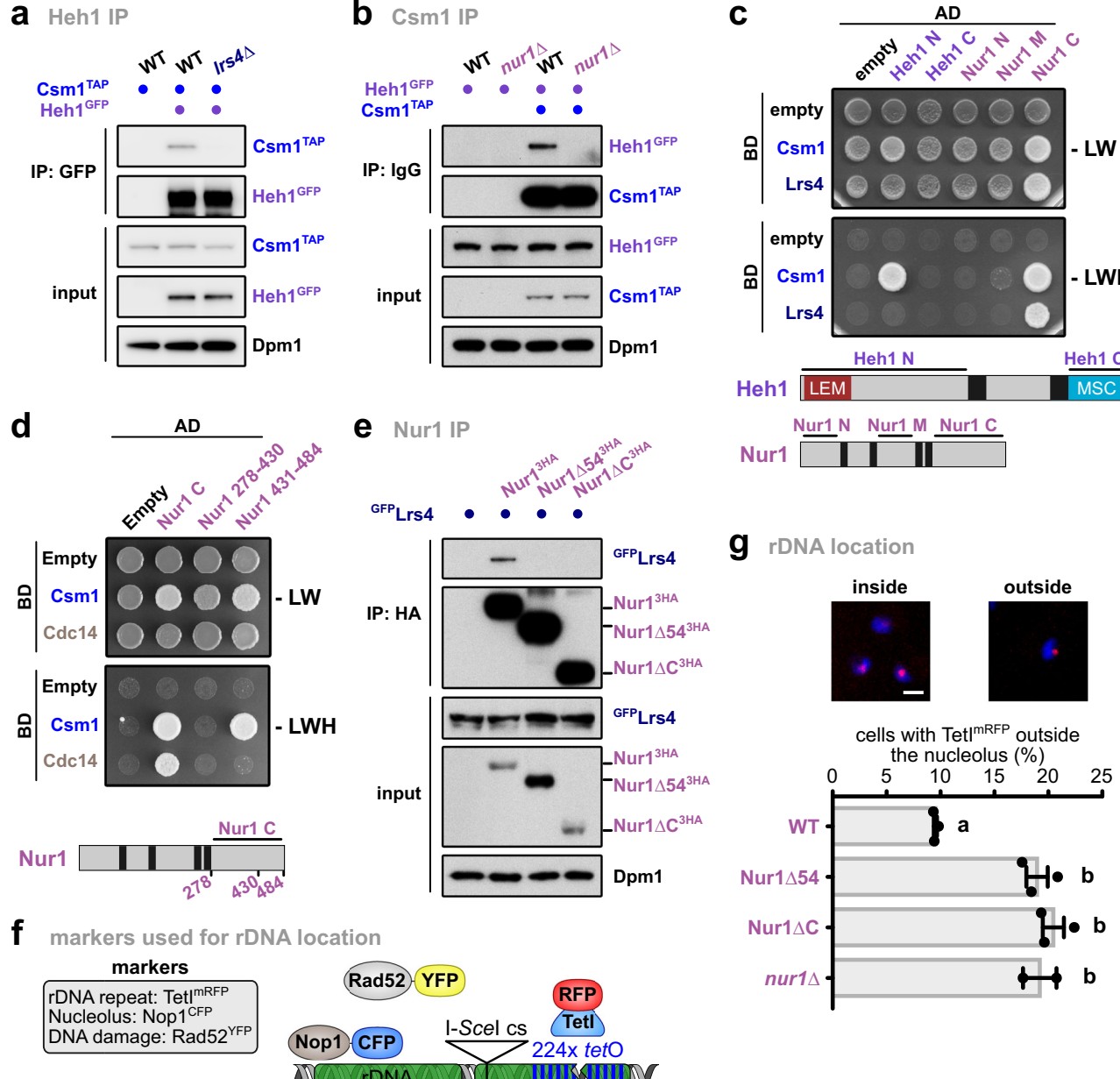

**Fig. 2 The Nur1 C-terminus is critical for CLIP-cohibin interaction.** Co-immunoprecipitation of Csm1[TAP] and Heh1[GFP] in WT, *lrs4Δ* (**a**) and *nur1Δ* (**b**) cells. Dpm1 served as loading control. Y2H analysis of Csm1 and Lrs4 with Heh1 and Nur1 nucleoplasmic domains (**c**); or of Csm1 and Cdc14 with Nur1 C and truncations (**d**). Fusions with Gal4-activating domain (AD) or Gal4-DNA-binding domain (BD) are indicated. Scheme of the constructs used for CLIP complex components is shown. **e** Co-immunoprecipitation of [GFP]Lrs4 with either HA-tagged Nur1 full-length (Nur1[3HA]), lacking its last 54 residues (Nur1Δ54[3HA]) or the complete C-terminal domain (Nur1ΔC[3HA]). Dpm1 served as loading control. **f** Scheme of rDNA locus and fluorescent markers used in **g**. Cells bear a *tetO* array adjacent to an I-*Sce*I endonuclease cut site inserted into an rDNA unit on chromosome XII, which is revealed by TetI[mRFP] foci. These cells also express Rad52[YFP] as a marker of the HR machinery. The nucleolus is visualized by a plasmid expressing the nucleolar protein Nop1[ECFP]. **g** Percentage of undamaged Nur16HA, Nur1Δ54[6HA], Nur1ΔC[6HA], or *nur1Δ* cells with rDNA repeats localized outside the nucleolus. Repeat location was monitored by the position of TetI[mRFP] focus relative to the nucleolar mark, and quantification and representative images are shown. Data are mean ± SEM of $n = 3$ independent biological replicates or $n = 2$ for *nur1Δ*. Scale bar, 2 μm. Statistical analysis was performed using analysis of variance (ANOVA), and letters denote significant differences with a Tukey's post hoc test at $P \leq 0.05$. Source data are provided as a Source Data file.

that Nur1 interaction with Csm1 was abolished when residues S441, T446, and S449 were mutated to aspartate, but unaffected when replaced by alanine (Supplementary Fig. 3d). We further confirmed the critical role of these residues for the CLIP-cohibin interaction in vivo using coIP (Fig. 3b).

Lack of perinuclear tethering promotes nucleolar release and Rad52-mediated recombination of rDNA repeats[6], which can be quantitated by the loss of an *ADE2* reporter gene inserted into the

rDNA locus. Appearance of half-sectored colonies indicates marker loss by recombination, which most likely occurs through unequal sister-chromatid exchange (USCE; see "Methods" section). When assessing the rate of marker loss in the *nur1* phosphomimetic mutant, we found increased rDNA recombination similar to the *nur1Δ* mutant[6]. This indicates that Nur1 phosphorylation decreases rDNA stability (Fig. 3c). These results imply that the CLIP-cohibin association is dynamically controlled

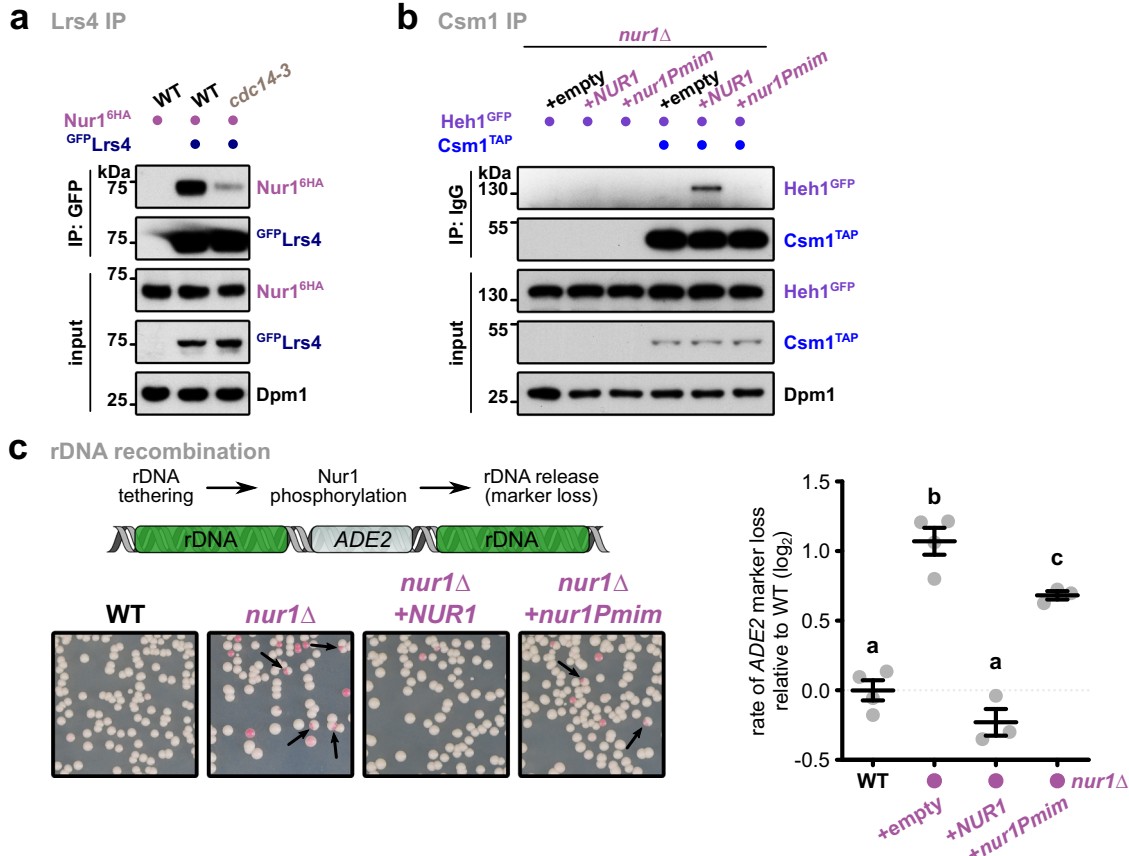

**Fig. 3 C-terminal Nur1 phosphorylation disrupts the rDNA tethering complex. a** Co-immunoprecipitation of Nur1$^{6HA}$ with $^{GFP}$Lrs4 in WT or *cdc14-3* mutant cells. Cells were grown at the permissive temperature and shifted to 37 °C for 1 h. Dpm1 served as loading control. **b** Co-immunoprecipitation of Heh1$^{GFP}$ with Csm1$^{TAP}$ in *nur1Δ* cells. The strains were transformed with empty vector or plasmids bearing *NUR1* or its phosphomimetic mutant (*nur1Pmim*) expressed from the endogenous promoter. Dpm1 served as loading control. **c** Quantification of rDNA recombination rates in WT or *nur1Δ* cells as measured by marker loss of an *ADE2* marker inserted into rDNA. Cells have been transformed with empty vector or plasmids bearing *NUR1* or *nur1Pmim* expressed from the endogenous promoter. The rate of marker loss is calculated as the ratio of half-sectored colonies (as indicated by the arrows) to the total number of colonies, excluding completely red colonies. Data are mean ± SEM of *n* independent biological replicates (*n* = 4 for WT and *nur1Δ*; *n* = 3 for *NUR1* and *nur1Pmim*), shown in log$_2$ scale relative to WT. Statistical analysis was performed using ANOVA, and letters denote significant differences with a Tukey's post hoc test at $P \leq 0.05$. Source data are provided as a Source Data file.

by Nur1 phosphorylation and counterbalanced by the phosphatase Cdc14. While the high-copy number of rDNA repeats inside the nucleolus may favor HR and marker loss, Rad52-mediated recombination is suppressed by SUMOylation and Pol I transcription[21,22]. In agreement with a previous study[33], we found that the absence of Uaf30 (upstream activation factor 30), which impairs Pol I transcription, resulted in rDNA instability (Supplementary Fig. 3e). Interestingly, while rDNA recombination in the *uaf30Δ* mutant is higher than in *csm1Δ* or *nur1Δ* cells, combined deletions resulted in a reduced rate of marker loss similar to the *csm1Δ* single mutant. This result suggests that 'fast' recombination in the *uaf30Δ* mutant is by-passed by a 'slower' recombination rate when rDNA tethering is compromised, further supporting the notion that loss of CLIP or cohibin promotes the nucleolar release of broken rDNA repeats into the nucleoplasm. Since only a fraction of rDNA repeats is likely released, we surmise that the lower recombination rate is due to the reduced number of extranucleolar repeats that can serve as donor sequences during HR.

**CLIP-cohibin dissociation depends on SUMOylation.** SUMOylation plays a prominent role in the DNA damage response and rDNA stability[9] and promotes the relocation of

genomic regions after damage[12–17]. We therefore examined whether this modification also facilitates rDNA release from the nucleolus. Since SUMO targets may be not or only partially modified in the absence of DNA damage, we decided to over-express SUMO to increase the SUMOylation steady-state levels (Supplementary Fig. 4a). Testing the CLIP-cohibin interaction, we found that SUMO overexpression decreased Nur1 association with Lrs4 but not with its partner Heh1 (Fig. 4a, Supplementary Fig. 4a). We further found that SUMO overexpression results in high *ADE2* marker loss, indicating decreased rDNA stability (Fig. 4b). Notably, additional deletion of *LRS4* resulted in a non-additive phenotype, suggesting that SUMOylation and Lrs4 act within the same pathway (Fig. 4b). Consistently, cells expressing the SUMO isopeptidase mutant *ulp2ΔC*, which lacks the region required for Csm1 interaction[34], also showed an increased rate of marker loss (Supplementary Fig. 4b). When studying the location of rDNA using an mRFP-marked repeat insertion, SUMO over-expression increased the number of cells with delocalized rDNA repeats (Fig. 4c), further supporting the role of SUMO in promoting rDNA release.

Several nucleolar proteins involved in rDNA maintenance are SUMOylated[34,35]. For instance, rDNA silencing is regulated through SUMO-dependent Tof2 degradation[34]. However, several of our results imply that SUMO overexpression-mediated rDNA

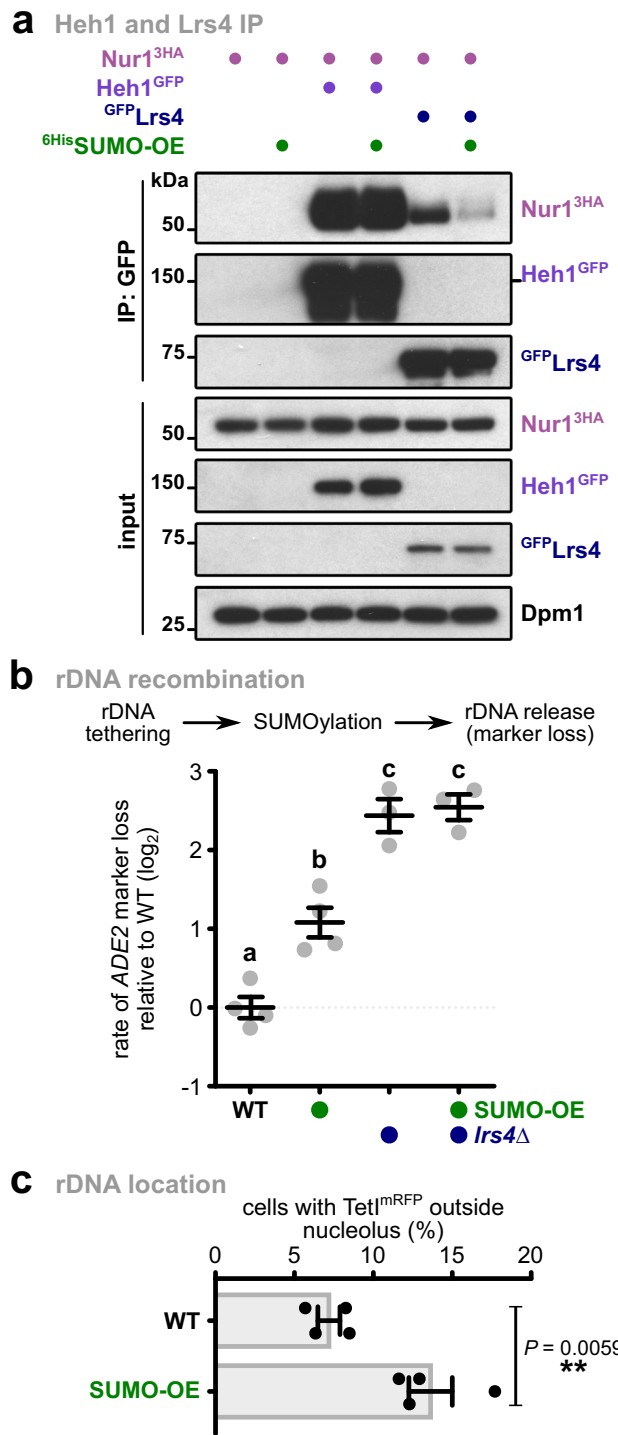

**a** Heh1 and Lrs4 IP

**b** rDNA recombination

**c** rDNA location

**Fig. 4 SUMO overexpression promotes CLIP-cohibin disassembly and rDNA release and recombination. a** Co-immunoprecipitation of Nur13HA with Heh1[GFP] or [GFP]Lrs4 in cells expressing SUMO at endogenous levels or overexpressed from the *ADH1* promoter ([6His]SUMO-OE). Dpm1 served as loading control. **b** Rates of rDNA recombination in WT ($n = 4$) or *lrs4Δ* ($n = 3$) cells expressing SUMO at endogenous levels or overexpressed from vectors with the *GAL1* promoter (SUMO-OE), grown in galactose-containing media for 4 days. Data are mean ± SEM of *n* independent biological replicates, shown in $\log_2$ scale relative to WT. Marker loss is calculated as in Fig. 3. Statistical analysis was performed using ANOVA, and letters denote significant differences with a Tukey's post hoc test at $P \leq 0.05$. **c** Percentage of undamaged cells with endogenous (WT) or overexpressed (SUMO-OE) SUMO levels with rDNA repeats localized outside the nucleolus, monitored as described for Fig. 2. Quantification of the marked rDNA unit was scored from four independent biological replicates, and the mean ± SEM is shown. Statistical analysis was performed using two-tailed Student's *t*-test. Source data are provided as a Source Data file.

Since both SUMOylation and Nur1 phosphorylation stimulate CLIP-cohibin dissociation, we tested whether SUMO overexpression induces Nur1 phosphorylation. To this end, we monitored the electrophoretic mobility of Nur1 using Phos-tag gels in cells with endogenous or elevated levels of SUMO. While Nur1 phosphorylation levels changed in a cell-cycle-dependent manner, in agreement with previous reports[30], SUMO overexpression did not affect these changes (Supplementary Fig. 4g). This result indicates that Nur1 phosphorylation is SUMO-independent, and either upstream or parallel to SUMOylation. Overall, these findings argue that SUMOylation disrupts the CLIP-cohibin complex, promoting the release of ribosomal repeats.

**Multiple members of the rDNA tethering complex are SUMOylated.** To obtain further insight into the mechanism triggering SUMO-dependent disruption of the CLIP-cohibin complex, we examined Nur1 and Lrs4 SUMOylation in vivo. Upon SUMO overexpression, Nur1 immunoprecipitation revealed an additional, slower migrating form that was further shifted when SUMO was fused to GFP (Fig. 5a). These modifications were absent in cells lacking the SUMO E3 ligase Siz2 (Fig. 5a). We further confirmed Nur1 SUMOylation by enriching for [6His]SUMO conjugates using Ni-NTA pulldowns under denaturing conditions (Fig. 5b). Interestingly, SUMOylated Nur1 was also reduced in *lrs4Δ* cells and *sir2Δ* cells, or when cells were treated with nicotinamide (Fig. 5b, Supplementary Fig. 5a, b), a strong inhibitor of Sir2 activity[36]. Together, these results imply that Nur1 SUMOylation is mediated by Siz2, and requires CLIP association with the rDNA locus, which depends on cohibin and likely Sir2[6,8]. Since previous studies failed to detect the SUMOylation of cohibin members[34,37], we overexpressed an isopeptidase-resistant mutant of SUMO (SUMO[Q95P]) to increase the detection sensitivity of SUMO conjugates[38]. Immunoprecipitation of Lrs4 under semi-denaturing conditions, which remove non-covalent interactions (see "methods" section), revealed an additional slower-migrating band recognized by an anti-SUMO antibody (Fig. 5c). We therefore conclude that Nur1 and Lrs4 are SUMOylated in vivo, which is in line with two proteomics studies[39,40].

To examine whether SUMOylation of the CLIP-cohibin complex affects rDNA stability, we determined the SUMO target sites on Nur1 by mutating putative acceptor lysine residues (K) to arginine (R). Using Ni-NTA pulldowns with [6His]SUMO and IP assays, we found that combined mutation of the acceptor lysine residues K175 and K176 abolished SUMOylation of Nur1

release is unrelated to Tof2 SUMOylation and rDNA silencing. SUMO overexpression did not affect transcript levels of the non-transcribed spacers NTS1 and NTS2 (Supplementary Fig. 4c). Whereas RENT and Tof2 binding to NTS1 is impaired in *ulp2Δ* cells[35], SUMO overexpression did not affect their enrichment at the repeats (Supplementary Fig. 4d). Tof2 protein levels also remained stable in SUMO overexpressing strains (Supplementary Fig. 4e). Furthermore, Csm1 association with NTS1 was not reduced, but instead enhanced under these conditions (Supplementary Fig. 4f). Together, these results suggest that SUMO-mediated rDNA release does not involve dissociation of RENT or Tof2 from the repeats and that Csm1 remains bound to rDNA after CLIP dissociation.

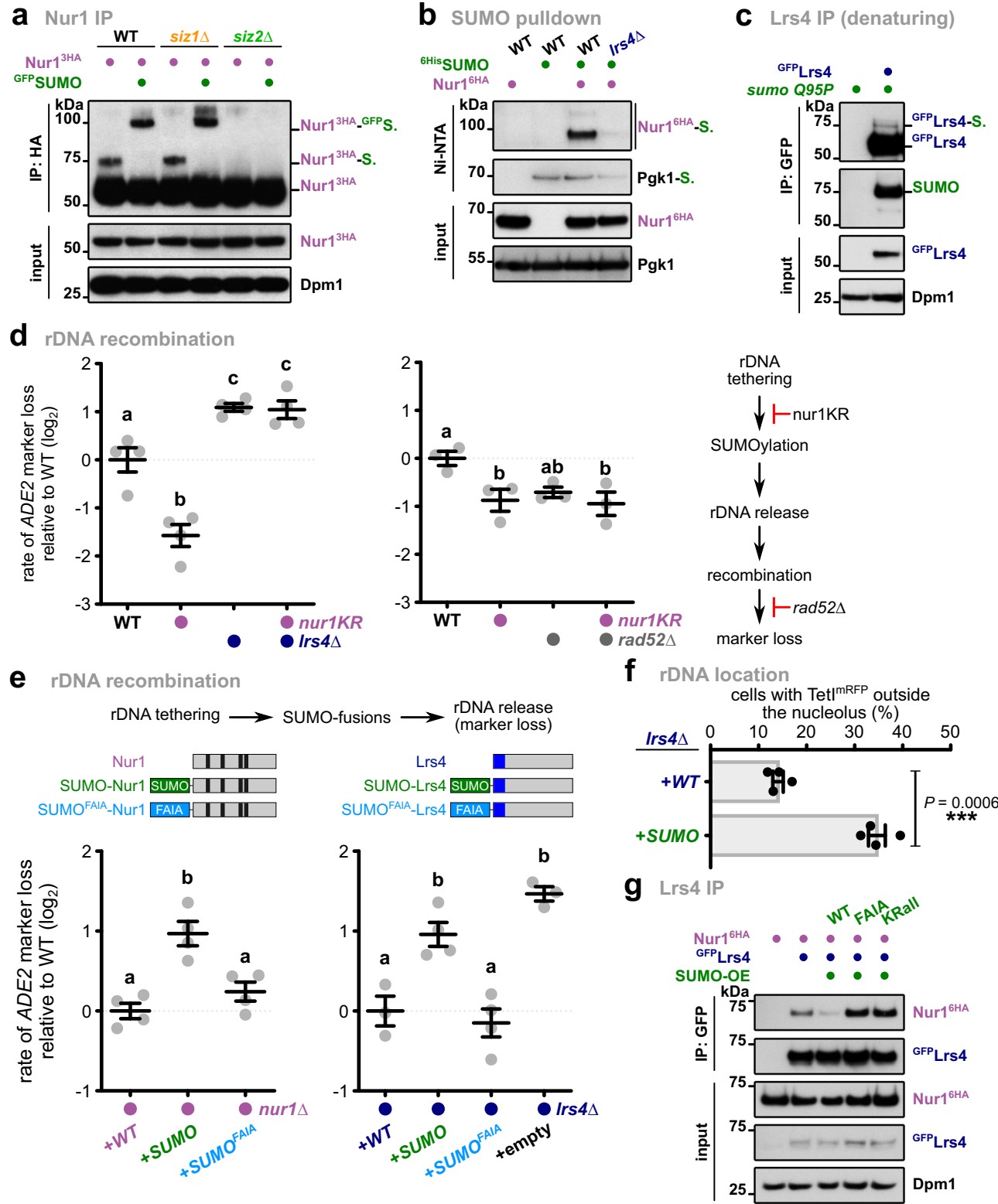

(Supplementary Fig. 5c, d). Notably, replacing wild-type Nur1 with this SUMO-deficient mutant (*nur1KR*) resulted in decreased rDNA marker loss, which was entirely dependent on the presence of Lrs4 (Fig. 5d, left). Moreover, while *RAD52* deletion decreases recombination, as previously reported[21], additional mutation of SUMOylation acceptor sites in Nur1 did not further reduce marker loss in *rad52Δ* cells (Fig. 5d, right). Overexpression of SUMO partially alleviates the recombination phenotype of

*nur1KR* (Supplementary Fig. 5e, left). To further validate the effect of SUMO on rDNA stability, we generated strains expressing Nur1 fused to the SUMO protease domain of Ulp1 (Nur1-UD). This fusion is expected to reduce SUMOylation of the rDNA tethering complex[41]. Notably, the presence of Nur1-UD decreased the rates of marker loss compared to the fusion with the catalytically dead domain of Ulp1 (Nur1-uDi; Supplementary Fig. 5e, right). Together, this argues that Nur1

**Fig. 5 SUMOylation of Nur1 and Lrs4 triggers CLIP-cohibin dissociation. a** Immunoprecipitation of Nur1[3HA] in WT, *siz1Δ*, and *siz2Δ* expressing SUMO (endogenous promoter) or [GFP]SUMO (*ADH1* promoter). Bands corresponding to Nur1 unmodified or monoSUMOylated are labeled. Dpm1 served as loading control. **b** Denaturing Ni-NTA pulldowns of [6His]SUMO conjugates from WT or *lrs4Δ* cells expressing Nur1[6HA], as indicated. Pgk1 served as control for loading and pulldown efficiency. **c** Immunoprecipitation under semi-denaturing conditions of [GFP]Lrs4 in cells overexpressing an isopeptidase-resistant SUMO mutant (SUMO[Q95P]). Detection of SUMO from immunoprecipitated samples is shown. Dpm1 served as loading control. **d** Rates of rDNA recombination in WT, *lrs4Δ* (left; *n* = 4) or *rad52Δ* (right; *n* = 3) cells expressing endogenously 6HA-tagged Nur1 or the SUMOylation deficient Nur1 K175-176R mutant (*nur1KR*). **e** Rates of rDNA recombination in *nur1Δ* or *lrs4Δ* cells transformed with empty vector or plasmids bearing 3HA-tagged *NUR1* (left; *n* = 4), TAP-tagged *LRS4* (right; *n* = 3 for *LRS4* and *lrs4Δ*; *n* = 4 for *SUMO-LRS4* and *SUMO[FAIA]-LRS4*) or the indicated linear fusions with the endogenous promoter. Used constructs for Nur1 and Lrs4 are shown. Black, predicted transmembrane domains; blue: Csm1-interacting region. **f** Percentage of *lrs4Δ* undamaged cells transformed with plasmids bearing TAP-tagged *LRS4* or *SUMO-LRS4* with rDNA repeats localized outside the nucleolus, monitored as described for Fig. 2. Quantification of the marked rDNA unit was scored from four independent biological replicates, and the mean ± SEM is shown. Statistical analysis was performed using two-tailed Student's *t*-test. **g** Co-immunoprecipitation of Nur1[6HA] with [GFP]Lrs4 in cells with endogenous SUMO levels, or strains overexpressing either SUMO WT or mutants unable to recognize SIMs (FAIA, which harbors the mutations F37A I39A) or to form SUMO chains (KRall). Dpm1 served as loading control. **d**, **e** The rate of marker loss is calculated as in Fig. 3; data are the mean ± SEM of *n* independent biological replicates, shown in log$_2$ scale relative to WT. Statistical analysis was performed using ANOVA, and letters denote significant differences with a Tukey's post hoc test at *P* ≤ 0.05. Source data are provided as a Source Data file.

SUMOylation promotes CLIP-cohibin dissociation and acts upstream of Rad52-mediated recombination. We further speculate that increased SUMOylation of other CLIP-cohibin members or other factors might compensate for the loss of Nur1 SUMOylation.

To validate the role of SUMOylation in the rDNA tethering complex through a gain-of-function approach, we expressed Nur1 and Lrs4 as linear fusions with SUMO in *nur1Δ* and *lrs4Δ* cells, respectively, and examined rDNA marker loss. Notably, unlike the WT versions, expression of Nur1 and Lrs4 fused to SUMO failed to complement the phenotype of the deletion mutants, resulting in increased marker loss similar to the empty vector (Fig. 5e). We consistently found that the expression of SUMO-Lrs4 increased the frequency of TetI[mRFP]-marked rDNA repeats outside the nucleolus (Fig. 5f). Together, these results suggest that the permanent presence of SUMO at Nur1 or Lrs4 disrupts the rDNA tethering complex. Since damaged ribosomal genes were shown to relocate to the nuclear periphery[20], we examined whether SUMO-Lrs4 expression alters the nuclear localization of rDNA repeats. Using a zoning assay[12,20], we determined the nuclear localization of TetI[mRFP] in cells expressing WT-Lrs4 and SUMO-Lrs4. In stark contrast to rDNA stability and nucleolar release (Fig. 5e, f), we did not observe a significant change in perinuclear localization for SUMO-Lrs4 (Supplementary Fig. 5f). This result implies that Lrs4 modification promotes untethering of rDNA repeats without affecting their recruitment to the nuclear periphery, which likely takes place after the nucleolar release.

Several proteins have been shown to interact with SUMO with moderate affinity through binding interfaces known as SIMs[10]. We therefore introduced mutations in the SUMO moiety (SUMO[FAIA]) in the linear Nur1 and Lrs4 fusion proteins preventing their recognition by SIM-containing proteins (Supplementary Fig 5g, h). When assessing rDNA marker loss, these SUMO[FAIA] fusions displayed recombination rates that are significantly lower than the corresponding WT-SUMO fusions and comparable to the non-fusion proteins (Fig. 5e). Moreover, we found that Lrs4 interaction with Nur1 is strongly reduced when WT SUMO is overexpressed, but not the SIM mutant (Fig. 5g). This implies that the specific recognition of SUMO by SIM-containing proteins, rather than the bulky presence of the SUMO moiety, is critical for rDNA tethering complex dissociation and rDNA recombination.

**Cdc48 recognizes the CLIP-cohibin complex and supports rDNA release.** The AAA-ATPase Cdc48 (also known as VCP/p97) acts as a segregase that disassembles protein complexes during proteasomal degradation, but also has functions in non-proteolytic pathways, such as DNA damage repair[42]. Besides recognizing ubiquitylated proteins, Cdc48 has been reported to bind to SUMOylated proteins[43,44]. Therefore, we tested whether the CLIP-cohibin complex is targeted by Cdc48 in a SUMO-dependent manner. CoIP demonstrates that Cdc48 interacts weakly with Lrs4 in WT cells. This interaction was increased in cells overexpressing a thermosensitive allele (*cdc48-6*) or an ATPase deficient mutant (*cdc48 E588Q*; Fig. 6a), known to block or delay the release of substrates, respectively[43,45]. We also observed increased Cdc48 interaction with Nur1 in *cdc48-6* cells (Fig. 6b). When examining Lrs4 binding to Nur1, their association was modestly increased in *cdc48-6* or *cdc48 E588Q* cells (Fig. 6c). Together, these results suggest that Cdc48 assists in the disruption of the rDNA tethering complex.

Cdc48 recognizes SUMOylated proteins through its substrate-recruiting co-factor Ufd1[43,44] that contains a SIM (Fig. 6d). Consistent with this, we found that Lrs4 and Nur1 interact with Ufd1 in Y2H assays in a SIM-dependent manner (Fig. 6e, f). Notably, *ufd1ΔSIM* cells displayed increased stability of the rDNA-inserted *ADE2* marker, suggesting that recognition of SUMOylated Nur1 or Lrs4 is critical for rDNA release and recombination. In agreement with this notion, reduced marker loss rates in *ufd1ΔSIM* mutant cells were not seen when Lrs4 was absent (Fig. 6g, left). Furthermore, SUMO overexpression did not significantly alter the recombination rate of *ufd1ΔSIM* cells (Fig. 6g, right). This implies that the role of Ufd1 in maintaining rDNA stability is specific to Lrs4 and occurs downstream of SUMOylation. We further tested the role of STUbLs, which have been reported to contribute to rDNA stability and recruit Cdc48 in a SUMO-dependent manner[34,35,46]. Interestingly, cells expressing a SUMO mutant lacking all lysine residues (SUMO[KRall])[37] showed increased Nur1 binding to Lrs4 and reduced marker loss (Fig. 5f, Supplementary Fig. S6a), suggesting a function of poly-SUMO or mixed SUMO-ubiquitin chains. However, deleting the STUbLs *SLX5*, *SLX8*, or *ULS1* did not affect Nur1 association to Lrs4 (Supplementary Fig. 6b, c). Together, these results suggest that Cdc48 recognizes both Lrs4 and Nur1, and promotes rDNA release in a SUMO- and Ufd1-dependent manner.

While our findings suggest that Nur1 phosphorylation does not depend on SUMOylation (Supplementary Fig. 4g), both modifications may contribute to rDNA release through the same pathway, with SUMOylation/Cdc48 acting downstream of Nur1 phosphorylation. We therefore assessed how the phosphorylated states of Nur1 affect rDNA recombination in cells deficient in

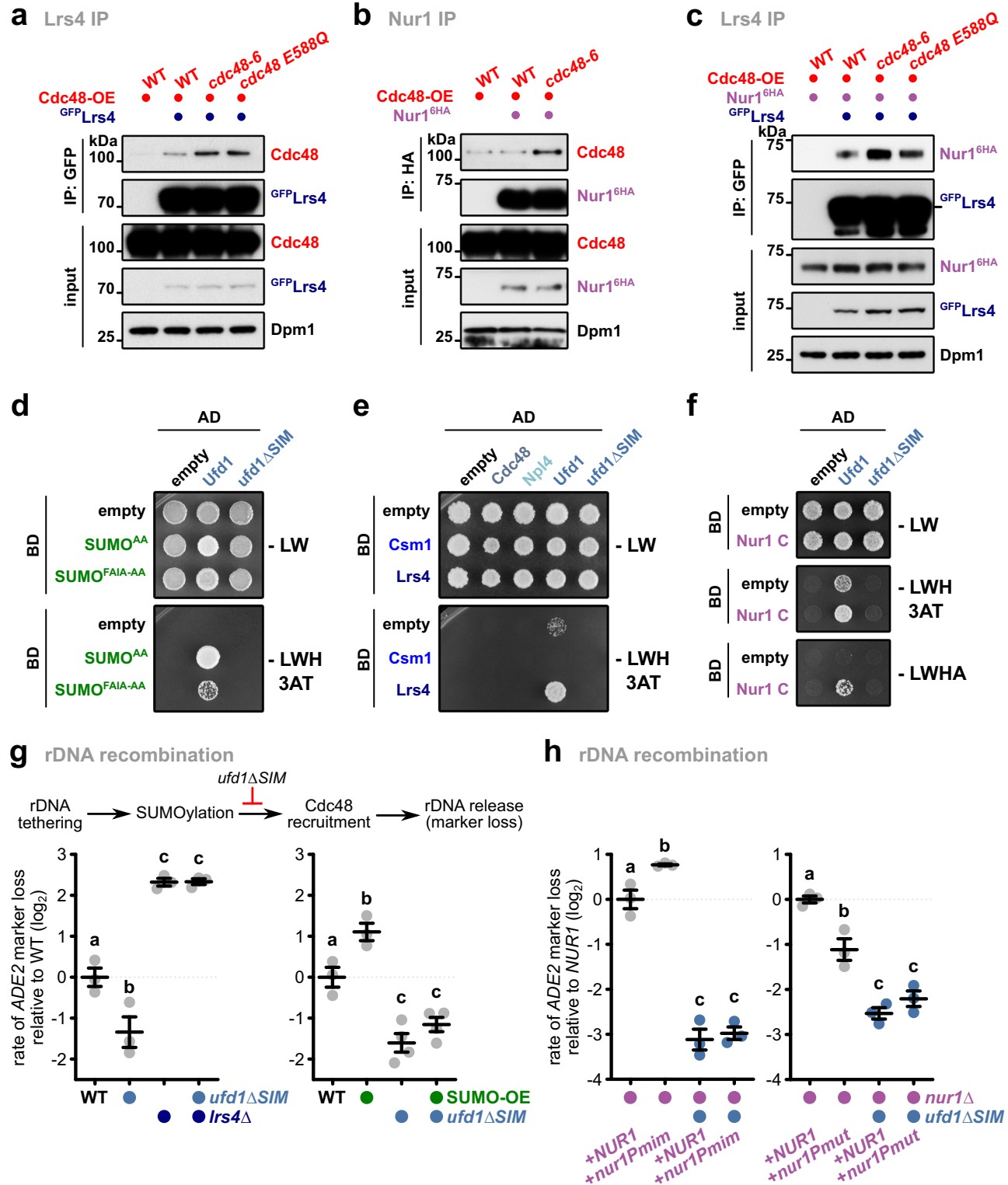

SUMO recognition. Notably, while expressing the phosphomimetic *nur1Pmim* mutant promoted rDNA recombination, this increase in marker loss was blocked when *nur1Pmim* was combined with *ufd1ΔSIM* (Fig. 6h, left). Conversely, cells deficient in Nur1 phosphorylation (*nur1Pmut*) that show reduced recombination rates displayed a non-additive phenotype in combination with the *ufd1ΔSIM* mutation (Fig. 6h, right). Together, these results suggest that SUMOylation acts downstream of Nur1 phosphorylation.

**DNA damage triggers the dissociation of the rDNA tethering complex.** DNA damage repair requires nuclear membrane release prior to the relocation of rDNA repeats outside the nucleolus. To examine whether CLIP-cohibin disassembly plays a role in this physiological context, we studied its association and modification in response to DSBs. To this end, we transiently expressed the *Physarum polycephalum* endonuclease I-*Ppo*I, which potentially generates DSBs at every rDNA repeat[47], using either the strong *GAL1* or the moderate *GALL* promoter. Quantitative PCR

**Fig. 6 SUMOylation recruits Cdc48 via its co-factor Ufd1 to assist in rDNA release.** Co-immunoprecipitation of Cdc48 with [GFP]Lrs4 (**a**) or Nur[16HA] (**b**), or of Nur[16HA] with [GFP]Lrs4 (**c**) in cells overexpressing WT or mutant Cdc48. Dpm1 served as loading control. **d** Y2H analysis of conjugation-deficient SUMO (SUMO[AA]) or the mutant unable to recognize SIMs (SUMO[FAIA-AA]) with Ufd1 and ufd1ΔSIM. **e** Y2H analysis of Csm1 and Lrs4 with Cdc48, Npl4, Ufd1, and ufd1ΔSIM. **f** Y2H analysis of Nur1 C with Ufd1 and ufd1ΔSIM. **g** Rates of rDNA recombination in WT, ufd1ΔSIM, lrs4Δ or ufd1ΔSIM lrs4Δ cells (left; n = 3), in WT and ufd1ΔSIM strains expressing SUMO at endogenous levels or overexpressed from vectors with the TEF1 promoter (SUMO-OE) (right; n = 3 for WT and SUMO-OE; n = 4 for ufd1ΔSIM and ufd1ΔSIM SUMO-OE). **h** Rates of rDNA marker loss in nur1Δ and nur1Δ ufd1ΔSIM cells transformed with plasmids bearing NUR1 or its phosphomimetic version (nur1Pmim) with the endogenous promoter (left; n = 3), or bearing NUR1 or its phosphomutant version (nur1Pmut) with the GAL1 promoter (right; n = 3). For nur1Pmut, cells were grown in galactose-containing media for 4 days. **d**–**f** Fusions with Gal4-activating domain (AD) or Gal4-DNA-binding domain (BD) are indicated. Cells spotted on -LWHA selective medium were grown for 4 days (**f**). The rate of marker loss is calculated as in Fig. 3; data are mean ± SEM of n independent biological replicates, shown in log$_2$ scale relative to WT (**g**) or nur1Δ+NUR1 (**h**). Statistical analysis was performed using ANOVA, and letters denote significant differences with a Tukey's post hoc test at $P \leq 0.05$. Source data are provided as a Source Data file.

analysis of genomic DNA confirmed that I-PpoI expression induces DSBs within rDNA repeats in a dose-dependent manner, without affecting rDNA copy number (Fig. 7a, b). We found that Heh1 binding to Lrs4 was reduced upon I-PpoI induction (Fig. 7a), whereas Lrs4 association with rDNA was not altered (Fig. 7b), suggesting that rDNA breaks trigger CLIP-cohibin disassembly. In agreement, we observed that I-PpoI induction increased Nur1 phosphorylation and the appearance of global SUMO conjugates (Supplementary Fig. 7a, b). To investigate whether preventing CLIP-cohibin disassembly affects rDNA release after damage, we monitored the nucleolar localization of tetO/TetI[mRFP]-marked repeats flanked by a single I-SceI endonuclease cut site. As previously reported[21], induction of I-SceI caused TetI[mRFP]-marked rDNA movement to an extranucleolar site (Fig. 7c), which is accompanied by an increase in Rad52[YFP] foci that often co-colocalize with TetI[mRFP] (Supplementary Fig. 7c). Notably, nucleolar rDNA release was substantially reduced in a ufd1ΔSIM mutant (Fig. 7c, top). A similar albeit less pronounced behavior was observed for nur1KR cells (Fig. 7c, bottom). Conversely, extranucleolar TetI[mRFP] foci were not further increased compared to WT cells when rDNA damage was induced in the C-terminally Nur1Δ54 mutant deficient in Lrs4 interaction (Fig. 7c, bottom). Similar to I-PpoI, spontaneous DSBs and rDNA recombination can be induced by mutants of the helicase Rrm3 and the replisome component Ctf4, which normally block DSBs formation at arrested replication forks[48,49]. Notably, expression of ufd1ΔSIM or nur1KR largely suppressed the recombination phenotype of rrm3Δ or ctf4Δ cells (Fig. 7d), suggesting that nucleolar rDNA release is important to repair DSBs generated by replication fork stalling. Based on these results, we conclude that DSBs at the rDNA repeats promote nucleolar release through controlled disassembly of the tethering complex through Nur1 phosphorylation and SUMO-dependent Cdc48 activity.

**UFD1L promotes rDNA release in human cells upon DNA damage.** Since nucleolar segregation of ribosomal repeats during DNA damage repair is a conserved mechanism[23,24], we speculated that key processes regulating rDNA dynamics might also be present in humans. To test this, we examined rDNA release upon damage in RPE (retinal pigment epithelium) cells depleted of the Ufd1 homolog, UFD1L. To generate DSBs in rDNA repeats, we used a tetracyclin-inducible expression system consisting of I-PpoI fused to the FKBP12 destabilization domain, which can be stabilized by the addition of the compound shield-1[50]. Release of damaged repeats from the nucleolus was assessed by quantifying the location of γH2AX foci, a marker for damaged DNA sites, relative to the nucleolar marker NPM (nucleophosmin). As previously reported[23,24,50], broken repeats were released to a large extent from the nucleolus of control cells upon expression of I-PpoI, as calculated by the number of cells with γH2AX foci within

the nucleolus (Fig. 8a, b). Upon depletion of UFD1L, we found that nucleolar segregation was reduced by more than 50% after damage relative to inhibition of the DNA damage checkpoint kinase ATM (ataxia telangiectasia-mutated), which resulted in a complete block of nucleolar release (Fig. 8a–c). Noteworthy, direct targeting of p97 to SUMOylated targets has not yet been described in humans, in contrast to yeast. However, p97 is targeted to SUMOylated substrates with the aid of the STUbL RNF4[42,51]. Notably, depletion of RNF4 also blocked rDNA nucleolar release upon damage, and additional UFD1L knockdown resulted in a non-additive phenotype, suggesting that RNF4 and p97 together assist nucleolar segregation (Fig. 8b). These results suggest that the role of Cdc48/p97 in controlling rDNA release during rDNA damage repair is conserved from yeast to humans.

## Discussion

The nucleolus shields the rDNA locus from uncontrolled recombination in eukaryotes, but this requires relocation to the nucleoplasm when DNA damage repair is needed[21,24]. However, how individual rDNA repeats exit the nucleolus was previously poorly understood. Here, we have identified the critical steps triggering the nucleolar release of rDNA repeats in yeast both under normal growth conditions and after exogenous DNA damage. These molecular events include Nur1 phosphorylation (Fig. 3), which accumulates upon DSB formation in rDNA repeats (Supplementary Fig. 7). It further requires Siz2-dependent SUMOylation of Nur1 and Lrs4, which occurs downstream of Nur1 phosphorylation and earmarks the rDNA tethering complex for disassembly and occurs downstream of Nur1 phosphorylation (Figs. 5 and 6). Repeat release from the membrane-bound CLIP complex is ultimately mediated by the AAA-ATPase Cdc48/p97 in conjunction with its cofactor Ufd1 through recognition of SUMOylated Nur1 and Lrs4, which allows rDNA relocation and repair by Rad52 (Figs. 6 and 7).

The segregase Cdc48/p97 plays a key role in many cellular processes, including the DNA damage response, where it segregates ubiquitylated proteins from non-modified partner proteins to facilitate their proteasomal degradation. Recognition of ubiquitylated proteins is mediated by the heterodimeric cofactor Ufd1-Npl4, whereas other co-factors help Cdc48 to orchestrate the delivery to the proteasome[42]. However, several studies revealed that Cdc48 also participates in disassembling protein complexes marked with SUMO[43,44,52]. Consistent with these reports, we discovered that Cdc48 binds to SUMOylated Nur1 and Lrs4 through Ufd1 and disrupts the CLIP-cohibin complex to promote rDNA relocation upon damage (Figs. 6 and 7). Presenting multiple SUMOylated substrates may increase the efficiency of Cdc48[Ufd1-Npl4]-mediated CLIP-cohibin disassembly, and in agreement, we found that overexpressing SUMO compensates for deficient rDNA recombination in the nur1KR

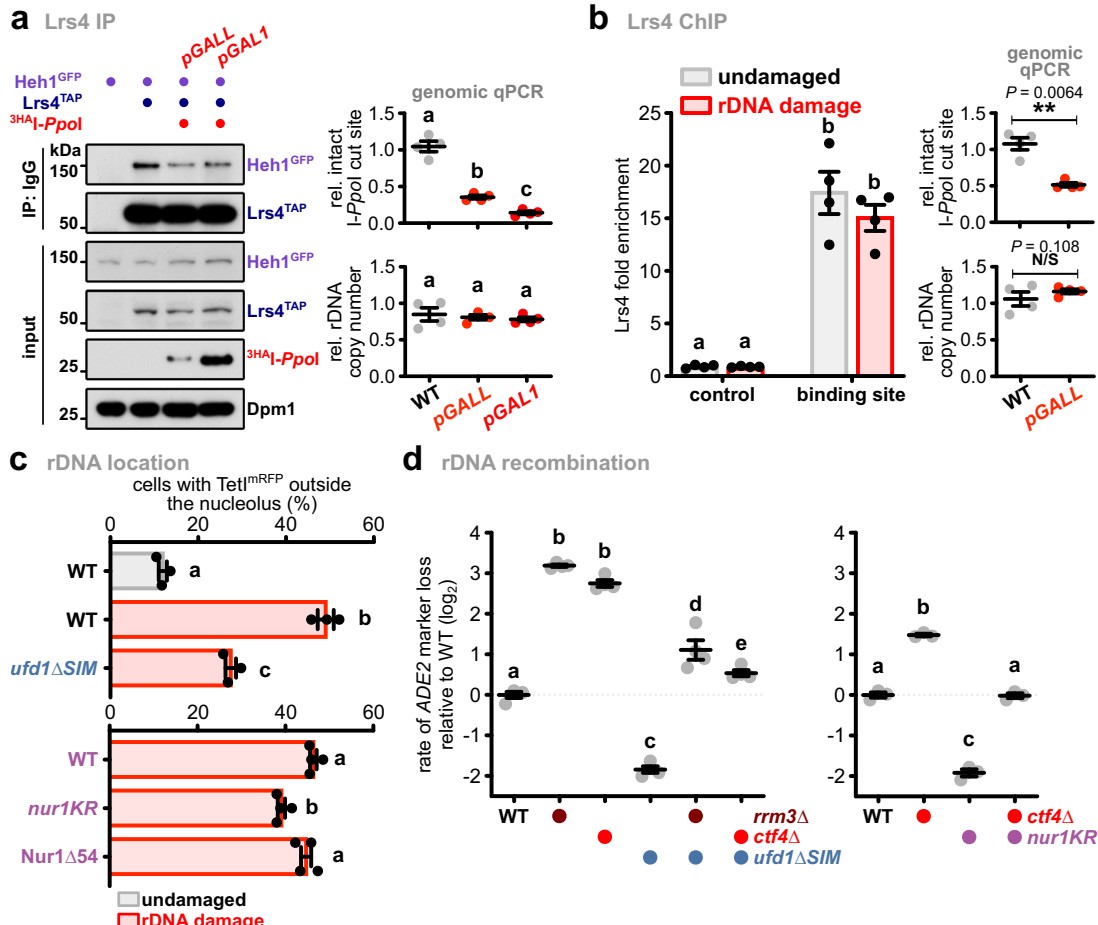

**Fig. 7 DNA damage contributes to CLIP-cohibin disassembly and Cdc48-dependent nucleolar rDNA release. a** Co-immunoprecipitation of Heh1[GFP] with Lrs4[TAP] in cells with or without rDNA damage generated by [3HA]I-*Ppo*I expression. Dpm1 served as loading control. Quantification of double-strand break (DSB) induction (top right) and rDNA copy number (bottom right) relative to undamaged cells is shown ($n = 4$ independent biological experiments). **b** Lrs4[TAP] binding to rDNA repeats in cells with or without rDNA damage, quantified by ChIP-qPCR ($n = 4$ independent biological experiments). The ChIP values are shown as Lrs4 fold enrichment over the average of three rDNA positions, after normalization to input. Quantification of double-strand break (DSB) induction (top right) and rDNA copy number (bottom right) relative to undamaged cells is shown ($n = 4$ independent biological experiments), and the statistical analysis was performed using two-tailed Student's *t*-test. **c** Percentage of WT and *ufd1ΔSIM* cells (top; $n = 3$) or expressing Nur1 full-length (WT), the SUMOylation deficient mutant (*nur1KR*) or lacking its last 54 residues (Nur1Δ54) (bottom; $n = 4$), with rDNA repeats localized outside the nucleolus before and after DSB induction, quantified as in Fig. 2. **d** Quantification of rDNA recombination rates in the indicated strains (left, $n = 4$; right, $n = 3$). The loss of an *ADE2* marker inserted into rDNA is calculated as in Fig. 3, and shown in log$_2$ scale relative to WT. **c, d** Data are mean ± SEM of *n* biologically independent experiments. Statistical analysis was performed using ANOVA, and letters denote significant differences with a Tukey's post hoc test at $P \leq 0.05$. Source data are provided as a Source Data file.

mutant (Supplementary Fig. 5). In addition, we found that depletion of the p97 cofactor UFD1L impairs the nucleolar segregation of broken rDNA repeats in human cells (Fig. 8). This implies that the role of Cdc48/p97[Ufd1-Npl4] in rDNA release is conserved between yeast and humans, although the human substrate(s) remain(s) to be identified. Given the presence of mixed SUMO-ubiquitin chains, generated by STUbL E3 ligases, it has been proposed that Cdc48 can be recruited to its targets by a dual mechanism employing the SUMO- and ubiquitin-specific binding moieties of Ufd1 and Npl4, respectively[42,44]. We found that deletion of the yeast STUbLs did not affect CLIP-cohibin dissociation (Supplementary Fig. 6). However, we cannot exclude the possibility that STUbLs are involved downstream of CLIP-cohibin disassembly or act redundantly. It is thus noteworthy that the human STUbL RNF4 acts together with UFD1L to release rDNA from the nucleolus upon damage (Fig. 8), implying a functional link between ubiquitylation and p97 in relocating rDNA in human cells.

In addition to SUMO modification, Nur1 phosphorylation is another critical step controlling rDNA release. Specifically, we found that a mutant mimicking the phosphorylated state of Nur1 promotes DNA repair (Fig. 3), whereas the corresponding mutant deficient in phosphorylation suppresses recombination (Fig. 6). Nur1 is phosphorylated in a cell cycle-dependent manner by the cyclin-dependent kinase Cdk1, which is counteracted by the phosphatase Cdc14[30,32]. Interestingly, phosphorylated Nur1 inhibits Cdc14 activation, and promotes Cdc14 release from the nucleolus once dephosphorylated in early anaphase[30]. We found that Nur1 phosphorylation is induced in response to DSBs (Supplementary Fig. 7a). While CDK is inhibited after DNA damage in metazoans[53], yeast Cdk1 was previously reported to be active at DSBs[54–56]. It is therefore possible that this kinase is also involved in Nur1 phosphorylation during rDNA damage. Furthermore, cell cycle-dependent Nur1 phosphorylation might also serve as a signal to release the rDNA locus from the nuclear envelope to

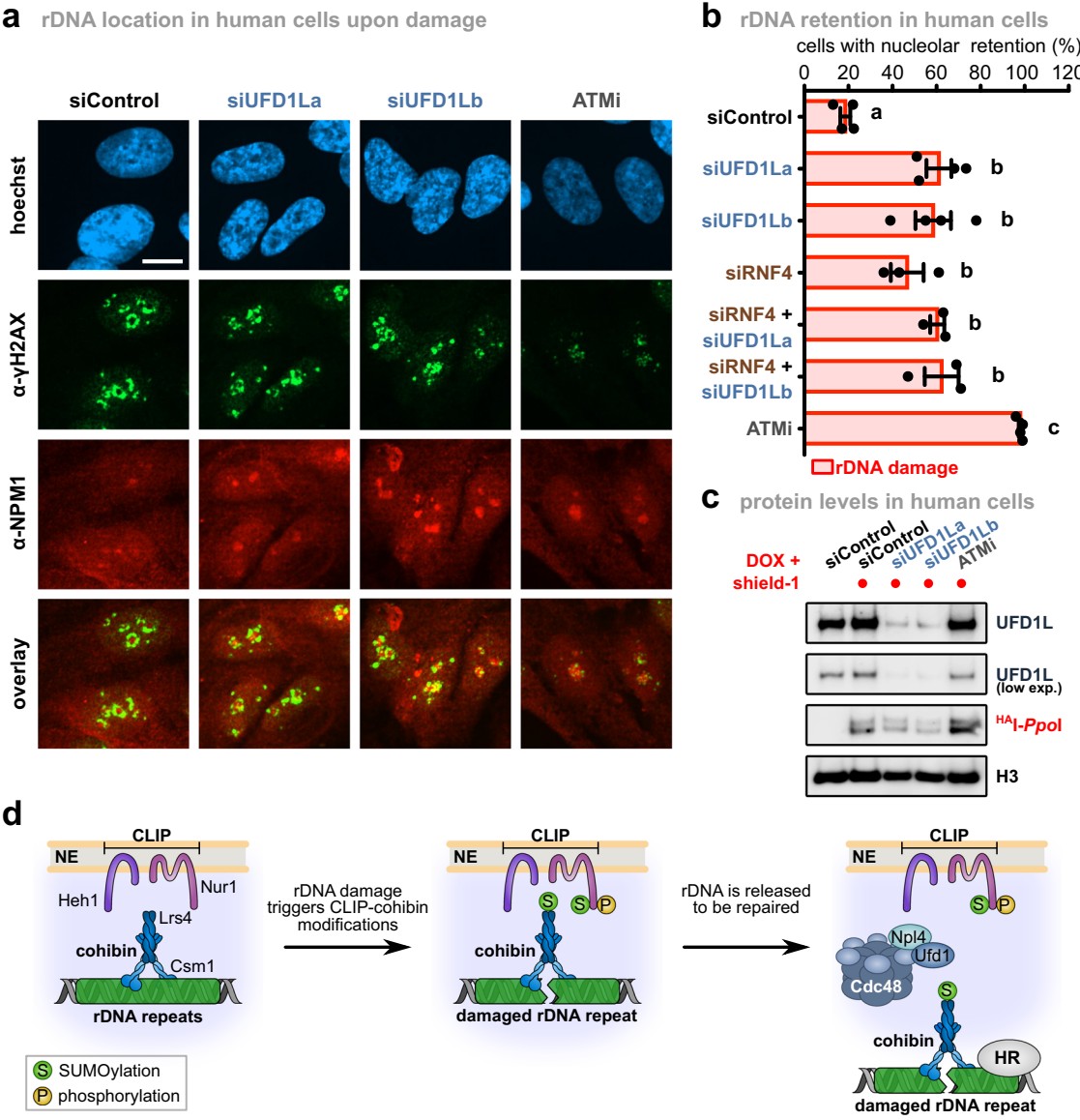

**Fig. 8 Nucleolar segregation of broken repeats requires UFD1L in human cells. a** Retention of rDNA locus (NPM1) in human cells upon DNA damage treated with siRNA against UFD1L. Human RPE cells stably expressing FKBP12-[HA]I-*Ppo*I were transfected with siRNAs against UFD1L (siUFD1La and siUFD1Lb), or treated with the ATM kinase inhibitor KU55933 (ATMi). Representative images of immunofluorescence staining after rDNA damage, using the indicated antibodies, are shown. Scale bar, 10 μm. **b** Quantification of human RPE cells transfected with siRNA against a control (siControl; $n = 4$), RNF4 (siRNF4; $n = 3$), siUFD1L (siUFD1La and siUFD1Lb; $n = 4$), combination thereof ($n = 3$), or ATMi treated ($n = 4$), showing the percentage of cells with nucleolar retention of γH2AX foci. Data are mean ± SEM of $n$ biologically independent experiments. Statistical analysis was performed using ANOVA, and letters denote significant differences with a Tukey's post hoc test at $P \leq 0.05$. **c** Immunoblot of RPE cells from Fig. 7a after rDNA damage, showing knock-down efficiency of UFD1L and [HA]I-*Ppo*I induction. H3 served as loading control. **d** Model for rDNA release upon DNA damage: Under normal conditions, the rDNA repeats are kept inside the nucleolus by Cdc14-mediated dephosphorylation of Nur1, which maintains the interaction of the rDNA tethering complex. When a repeat unit is damaged, SUMOylation of CLIP-cohibin disrupts the complex through Cdc48-Npl4-Ufd1, allowing rDNA relocation outside the nucleolus. Nur1 is also phosphorylated, further supporting the broken repeat relocation. Source data are provided as a Source Data file.

allow chromosome segregation, which would be counteracted by Cdc14 activity at the end of mitosis[30,57]. Alternatively, Nur1 could also be modified by DNA damage checkpoint kinases. For instance, ATM-mediated phosphorylation of Treacle (also known as *TCOF1*) modulates the nucleolar response during rDNA damage in humans[58,59]. The molecular mechanism by which Nur1 phosphorylation triggers CLIP-cohibin disruption remains unknown. While SUMOylation does not affect Nur1 phosphorylation (Supplementary Fig. 4), both appear to act in the same pathway (Fig. 6), suggesting that Nur1 phosphorylation occurs upstream, and presumably controls SUMOylation.

Consistent with this model, phosphorylation of other targets is known to stimulate their SUMOylation[60].

Maintaining dynamic regulation of the CLIP-cohibin complex appears to be critical, since we find that tethering rDNA repeats permanently to the nuclear envelope is detrimental for cellular growth (Fig. 1, Supplementary Fig. 1). We favor the idea that this may be caused by the failure to detach rDNA repeats from the nuclear periphery during mitosis. Supporting this notion, release of the *S. japonicus* CLIP complex from centromeres and telomeres is crucial for proper progression of mitosis[61]. However, we cannot exclude the possibility that permanent tethering might

also affect the repair of rDNA repeats, which frequently undergo DNA damage and are found to be transiently released even in the absence of exogenous DNA damage[21]. Notably, artificial tethering of the mating-type loci in *S. cerevisiae*[62] or genomic regions recognized by Gal4 (Fig. 1c) allowed normal growth. Together, these data suggest that maintaining a dynamic spatial regulation is specifically important for repetitive sequences.

In conclusion, we propose a model where phosphorylation and SUMOylation of the rDNA tethering complex act together to promote its dissociation and modulate the release of broken repeats (Fig. 8d). This process likely occurs upstream of relocation of rDNA repeats to the nuclear periphery, as previously described[20]. The Smc5/6 subunit Nse3 was recently shown to control rDNA silencing and tethering[63], suggesting that additional factors might be involved in the regulation of rDNA release. Remarkably, the spatial control of damaged rDNA repeats is highly reminiscent of other repair pathways, specifically the repair of DSBs within pericentromeric heterochromatin in flies and mice[25,26]. In this case, HR progression is temporally blocked by SUMO and the Smc5/6 complex, whereas SUMOylation and ubiquitylation by STUbLs promote the relocation of the damaged locus outside heterochromatin, and its targeting to the nuclear periphery for DNA damage repair[15,25,26]. Our results provide another example of how cells utilize similar molecular mechanisms to control the relocation of repetitive sequences and thus maintain genome integrity.

## Methods

**Yeast strains**. Yeast strains isogenic to W303 or DF5 were used for genetic studies, and PJ69-7a, Y187 or Y2H Gold for Y2H assays. Full genotypes are listed in Supplementary Table 1.

**Yeast methods and molecular biology**. Standard protocols for transformation, mating, sporulation, and tetrad dissection were used for yeast manipulations. Unless indicated otherwise, cells were grown at 30 °C in YPD medium (1% yeast extract, 2% peptone, and 2% glucose) or synthetic complete medium lacking individual amino acids to maintain selection for transformed plasmids. Protein tagging and the construction of deletion mutants were conducted by a PCR-based strategy and confirmed by immunoblotting and PCR, respectively[64,65]. For galactose induction, cells were cultured in medium supplemented with 2% raffinose, and galactose was added to a mid-log phase yeast cells to a final concentration of 2%. Expression of Heh1 fusions, SUMO (Smt3) or Cdc48 was induced with galactose for 2 h 30′. To generate DNA damage at the rDNA locus, I-SceI and I-PpoI expression was induced with galactose for 2 h; for Lrs4 chromatin immunoprecipitation (ChIP), I-PpoI induction was performed for 1 h 30′. For Nur1 C-terminal tagging with the active or inactive catalytic domain of SUMO-protease Ulp1 were performed using the *pYM15-UD* and *pYM15-uD^i*, respectively[41]. For auxin-inducible degradation, cells expressing the protein of interest fused to an auxin-dependent degron sequence derived from IAA17 together with the F-box protein TIR1 from *Oriza sativa*[66] were cultured in YPD medium until mid-log phase, and the synthetic auxin analog 1-naphthaleneacetic acid was added to a final concentration of 1.5 mM for 2 h. Damage in rDNA repeats using I-PpoI fused to the mouse glucocorticoid receptor was induced by adding galactose and triamcinolone acetonide[67] to a final concentration of 2% and 7.5 or 75 μM, respectively, to a mid-log phase yeast cells growing in minimal selective medium with 2% raffinose for 3 h. Yeast growth assays were performed by spotting fivefold serial dilutions of the indicated strains on solid agar plates.

Cloning methods used standard protocols or the Gibson Assembly Master Mix (NEB). For Y2H experiments, the different constructs were cloned into *pGADT7* or *pGBKT7* vectors (Clontech™). Plasmids with point mutations or deletions were constructed by the PCR-based site-directed mutagenesis approach. All plasmids used in this study are listed in Supplementary Table 2. Maps DNA sequences are available upon request.

**Yeast two-hybrid assays**. The PJ69a strain was used to co-transform with the indicated plasmids, while the Y187 and Y2HGold strains were transformed with the indicated *pGBKT7* or *pGADT7* constructs, respectively, followed by mating. Spotting assays were performed on control medium (−LW) or selective medium with increased stringency (−LWH, −LWH + 1 mM 3-aminotriazol, −LWHA) and grown for 3 days, unless indicated otherwise.

**Immunoblotting**. Total protein extracts from $2 \times 10^7$ cells ($OD_{600} = 1$) were prepared by trichloroacetic acid (TCA) precipitation[65]. Proteins solubilized in HU

loading buffer (8 M urea, 5% SDS, 200 mM Tris-HCl pH 6.8, 20 mM dithiothreitol, and bromophenol blue 1.5 mM) were resolved on NuPAGE 4–12% gradient gels (Invitrogen), transferred onto polyvinylidene fluoride membranes (Immobilon-P) and analyzed by standard immunoblotting techniques using specific antibodies (see the "Antibodies" section).

**Co-immunoprecipitation**. Cell lysates were prepared by resuspending the pellets from 150 to 200 $OD_{600}$ units in 800 μl lysis buffer (50 mM Tris pH 7.5, 150 mM NaCl, 10% glycerol, 1 mM EDTA pH 8, 0.5% NP-40, 1x complete EDTA-free protease inhibitor cocktail (Roche), 2 mM PMSF, 20 mM N-ethylmaleimide). To block phosphatase activity, 1x tablet of PhosSTOP, 10 mM NaF and 20 mM ß-glycerophosphate were added to the lysis buffer. Cells were lysed by bead beating (Precellys 24, Bertin instruments) with zirconia/silica beads (BioSpec Inc.) and lysates were cleared by centrifugation (800 g, 5 min). Clarified extracts were incubated with pre-equilibrated antiHA-sepharose (Roche), GFP-Trap (Chromo-tek) or IgG-agarose beads (Sigma) for 1.5 h at 4 °C, beads were washed four times with lysis buffer and two times with wash buffer (50 mM Tris pH 7.5, 150 mM NaCl, 1 mM EDTA pH 8). Proteins were eluted by boiling with 30 μl HU loading buffer and analyzed by immunoblotting.

**Analysis of protein SUMOylation using semi-denaturing purification**. To examine SUMOylated GFP-tagged proteins, cell lysates from 200 $OD_{600}$ units from cells with or without rDNA damage were prepared as for coIP experiments. Clarified extracts were incubated with 12.5 μl pre-equilibrated GFP-Trap beads (Chromotek) for 1.5 h at 4 °C, beads were washed once with lysis buffer. As reported for the detection of other posttranslational modifications[68], the GFP-Trap beads were then washed five times with denaturation buffer (PBS, 8 M urea). The specifically bound proteins were eluted by boiling with 30 μl HU loading buffer and analyzed by immunoblotting.

**Analysis of protein SUMOylation by Ni-NTA pulldowns**. To detect proteins covalently modified with SUMO, 200 $OD_{600}$ units from cells expressing N-terminally histidine-tagged SUMO (six histidines) under control of the ADH1 promoter, integrated at the *URA3* locus, were collected. Subsequently, Ni-NTA pulldowns using Ni-NTA magnetic agarose beads (Qiagen) were largely performed as previously described[69]. Briefly, cell lysates were prepared by resuspending the pellets in lysis buffer (1.85 M NaOH, 7.5% β-mercaptoethanol) and protein precipitation using TCA. Pellets were washed two times with deionized $H_2O$, resuspended in Buffer A (6 M Guanidiniumchloride, 10 mM $NaH_2PO_4$-$H_2O$, 10 mM Tris, 0.05% Tween 20, pH 8), and incubated at room temperature for 2 h with agitation. After centrifugation, the supernatant was incubated overnight with 100 μl of Ni-NTA beads and imidazole 20 mM, washed three times with Buffer A with imidazole 20 mM and five times with Buffer C (8 M Urea, 100 mM $NaH_2PO_4$-$H_2O$, 10 mM Tris, 0.05% Tween 20, pH 6.3). Proteins were eluted by addition of 30 μl SDS 1% and incubation at 65 °C for 10 min. Elutes were resuspended in 30 μl HU loading buffer and analyzed by immunoblotting.

**Generation of a SIM-binding deficient SUMO variant**. To screen for SIM-binding deficient mutants, we utilized strains overexpressing non-deconjugatable SUMO (SUMO^Q95P) under control of the galactose inducible promoter *GAL1*, which decreased growth in WT cells. By contrast, SUMO^Q95P caused lethality in a strain lacking the SUMO-targeted ubiquitin ligase subunit Slx5 (*slx5Δ*). As Slx5 harbors multiple SIMs for substrate recognition, we assumed that mutating the SIM binding surface on SUMO^Q95P should cause lethality if expressed in WT cells, due to inhibited Slx5 interaction. In agreement to a previous report[70], a SIM-binding deficient mutation of the conserved F37 in SUMO still allowed growth when expressed as a non-deconjugatable variant (SUMO^FA,Q95P), suggesting that this SUMO variant still supports SUMO-SIM interaction. When we additionally mutated I39, WT cells expressing non-deconjugatable SUMO^FAIA,Q95P caused lethality, suggesting that Slx5 dependent clearance of this SUMO variant was blocked. The loss of SIM recognition was then assessed by coIP of Slx5 with GFP-tagged SUMO^Q95P or the SUMO^Q95P variant F37A I39A.

**Chromatin immunoprecipitation (ChIP)**. ChIP experiments were performed as described previously[71], with some modifications. Briefly, 150 ml cell cultures were grown to $OD_{600} = 0.8$–1.0, cross-linked (1% formaldehyde, 16 min, room temperature), and lysed by bead beating. The chromatin fraction was isolated and sheared to 200–500 bp fragments (30 min, 30 s on/off cycles, 90% amplitude) using a Q800R1 sonicator (QSonica). Immunoprecipitations were performed overnight at 4 °C with 100 μl of 25% slurry of prewashed IgG-agarose beads (Sigma) from lysates corresponding to 80–100 $OD_{600}$ of cells. The beads were washed and eluted, and the eluate was reverse cross-linked at 65 °C for 3 h and incubated with proteinase K for 2 h at 55 °C. DNA was cleaned up with ChIP DNA Clean & Concentrator™ kit (#D5201, Zymo Research). Immunoprecipitated DNA was quantified by qPCR using primaQUANT SYBR Master mix (Steinbrenner Labor-systeme GmbH. #SL-9902B) and a QuantStudio™ 3/5 Real-Time PCR system (Applied Biosystems/Thermo Fisher). Primers are listed in Supplementary Table 3. To avoid changes due to different rDNA copy number, the relative fold enrichment for each strain was determined over the average of three rDNA positions (RDN1

#9, RDN1 #12, and RDN1 #25), after normalization to input: [rDNA(IP)/rDNA (WCE)]/[mean(rDNA)(IP)/mean(rDNA)(WCE)]. Graphs were generated using Graphpad Prism and Inkscape.

**Determination of rDNA copy number and induction of double-strand breaks**. The amount of cells corresponding to $OD_{600} = 1$ ($2 \times 10^7$ cells) was harvested before and after induction of rDNA damage using I-*PpoI* (uninduced and induced samples, respectively), and genomic DNA was prepared using the MasterPure Yeast DNA Purification Kit (Epicenter). Genomic DNA was then used as input for qPCR with specific primers (Supplementary Table 3). The relative rDNA copy number or intact I-*PpoI* cut site (rDNA damage efficiency) was calculated as the ratio between induced and uninduced samples, after normalized to a control locus on chromosome VI (*ACT1*). Graphs were generated using Graphpad Prism and Inkscape.

**Quantitative RT-qPCR**. RT-qPCR analyses were performed as previously described[71]. Briefly, cells were lysed by bead beating (Precellys 24, Bertin instruments) using TRIzol reagent and zirconia/silica beads (BioSpec Inc.), followed by centrifugation at 13,000 rpm for 15 min at 4 °C. Recovered supernatant was extracted with chloroform and reprecipitated with isopropyl alcohol. Resuspended RNA was treated with DNaseI, and 10 mg of RNA was used in standard RT reactions using oligo[(dT)20-N] primers. cDNAs were quantified by qPCR using primaQUANT SYBR Master mix (Steinbrenner Laborsysteme GmbH) and a QuantStudio™ 3 or QuantStudio™ 5 Real-Time PCR system (Applied Biosystems/Thermo Fisher). Primers are listed in Supplementary Table 3. For the calculation of mean values and standard deviation from independent experiments, *ACT1*-normalized data sets are shown in $\log_2$ scale as relative to the mean value of the wild type. Graphs were generated using Graphpad Prism and Inkscape.

**Live-cell imaging and analysis**. For fluorescence microscopy, cells were diluted to $OD_{600} = 0.15$ in synthetic complete medium lacking individual amino acids, and grown until $OD_{600} = 0.8–1.2$ before imaging. Microscopy slides (MatTek) were pretreated with 1 mg ml$^{-1}$ concanavalin A solution. Imaging was performed on a Zeiss AxioObserver Z1 confocal spinning-disk microscope equipped with an Evolve 512 (Photometrics) EMM-CCD camera through a Zeiss Alpha Plan/Apo 100×/1.46 oil DIC M27 objective lens. Optical section images were obtained at focus intervals of 0.2 µm. Subsequent processing and analyses of the images were performed in Fiji/ImageJ. Maximum Z-projections were obtained, and contrast and brightness were adjusted for each individual image for optimal visibility. Only images showing a clear nucleolus staining and RFP dot were further analyzed and scored as follows: TetI$^{mRFP}$ foci either touching or fully overlapping with the Nop1$^{CFP}$ nucleolar envelope marker were scored as located at the periphery and inside the nucleolus, respectively, whereas all others were scored as outside the nucleolus. The zoning assay was performed as previously reported[12,20]. For induction of I-*SceI* endonuclease (rDNA damage), cells were grown at 30 °C in synthetic complete medium lacking individual amino acids plus 2% raffinose to an $OD_{600} = 0.6$ and expression of I-*SceI* was induced by addition of 2% galactose for 2 h.

**Quantification of rDNA marker loss by unequal sister-chromatid exchange assays**. Assays were performed as previously described[8], with some modifications. Cells were grown in YPDA or synthetic complete medium lacking individual amino acids to $OD_{600} = 0.8–1.2$, diluted 1:1,000 in distilled $H_2O$ and plated on synthetic complete medium plates. Cells were incubated at 30 °C (for 3 days in glucose-containing medium, or for 4 days in galactose-containing medium) and then transferred to 4 °C for 3 days to enhance color development. The rate of marker loss was calculated by dividing the number of half-red/half-white colonies by the total number of colonies. Red colonies were excluded from all calculations. Graphs were generated using Graphpad Prism and Inkscape.

**Cell culture**. RPE cells were grown in DMEM (Sigma) supplemented with Penicillin-Streptomycin (Life Technologies), L-glutamine (ThermoFisher) and 10% fetal bovine serum (FBS) (Takarabio) at 37 °C and 5% $CO_2$. RNAi experiments were performed using RNAiMax (Life Technologies) 2 days before treatments. The following siRNA sequences were used: siControl (5′-UGGUUUACAUGUCGAC UAA-3′; Metabion), siUFD1La (5′-GUGGCCACCUACUCCAAAUUU-3′; Metabion), and siUFD1Lb (5′-CUACAAAGAACCCGAAAGAUUUU-3′; Metabion). $^{HA}$I-*PpoI* expression was induced by adding doxycycline (1 µg/ml, Sigma) 4 h and Shield-1 (1 µM, Aobious) 1 h before fixation.

**Immunofluorescence and quantification of nucleolar release in human cells**. Cells were washed with PBS (Sigma) and fixed in 4% paraformaldehyde for 30 min at room temperature. Then, cells were permeabilized in CSK buffer (10 mM HEPES pH 7.4, 300 mM Sucrose, 100 mM NaCl, 3 mM $MgCl_2$) with 0.1% Triton-X for 30 min. Blocking was performed for 1 h with 5% FBS in PBS after which cells were stained in 1% FBS in PBS overnight at 4 °C. After primary staining, cells were washed three times 5 min with PBS and incubated with secondary antibodies for 1 h at RT. After another three 5 min washes in PBS, slides were mounted with Aqua-Poly/Mount (Polysciences). Images were obtained using an AxioObserver Z1

confocal spinning-disk microscope (Zeiss) equipped with an AxioCam HRm CCD camera (Zeiss) or a sCMOS ORCA Flash 4.0 camera (Hamamatsu) and a Plan/Apo 63 Å~/1.4 water-immersion objective. The percentage of cells with nucleolar retention of γH2AX foci were defined as cells that show an overlap (yellow) of the γH2AX (green) with the NPM (red) signal. Blinded visual scoring of rDNA damage retention was performed by two individuals. Graphs were generated using Graphpad Prism and Inkscape.

**Antibodies**. Polyclonal Smt3 (1:5,000), Slx5 (1:5,000), and Cdc48 (1:5,000) antibodies were raised in rabbits and have been described previously[72–74]. Monoclonal antibodies directed against the HA epitope (1:1,000; 3F10) and monoclonal antibody against GFP (1:1,000; B-2) were purchased from Roche and Santa Cruz Biotechnology, respectively. Mouse monoclonal antibodies against Dpm1 (1:2,000; 5C5A7) and Pgk1 (1:5,000; 22C5D8) were obtained from Invitrogen. Mouse monoclonal antibodies against phospho-H2AX (Ser139) (1:1,000, JBW301) were obtained from Merck Millipore. Rabbit polyclonal antibodies against Nucleophosmin (1:100, ab15440) and Rad53 (1:1000, ab104232) or peroxidase anti-peroxidase (1:1,000, p1291) were purchased from Abcam and Sigma-Aldrich, respectively. Secondary antibodies goat-α-mouse IgG Alexa fluor 488 (1:5,000; A11001) and donkey-α-rabbit IgG Alexa fluor 568 (1:5,000; A10042) were obtained from Thermo Fisher Scientific.

**Statistics and reproducibility**. Representative results of at least two independent experiments were presented in all of the figure panels for all blots. Analyses of the variance were performed, and pairwise differences were evaluated with Tukey's post hoc test using R statistical language (R Development Core Team, 2008); different groups are marked with letters at the 0.05 significance level. For all error bars, data are mean ± S.E.M. P values were generated using two-tailed Student's t-tests; N/S, $P \geq 0.05$, *$P \leq 0.05$, **$P \leq 0.01$, ***$P \leq 0.001$.

**Reporting summary**. Further information on experimental design and research design is available in the Nature Research Reporting Summary linked to this article.

## Data availability
The source data underlying Figs. 2a, b, e, g, 3a–c, 4a–c, 5a–g, 6a–c, g, h, 7a–d, and 8b, c and Supplementary Figs. 1b, e, f, 2a–c, e, 3a, b, e, 4a–g, 5a–f, 6a–c, and 7a–d are provided as Source Data files. All other relevant data are available from the authors upon request. Source data are provided with this paper.

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

## Acknowledgements

We thank members of the Braun and Ladurner labs for fruitful discussions; S. Lall from LSE editors for critical comments on the manuscript; R. H. Medema for RPE cell lines; J. Torres-Rosell for *pYM15-UD* and *pYM15-uD$^i$* plasmids; P. Bourilhon and D. Sinitski for mouse cDNA and reagents; S. Fischer-Burkart, T. S. van Emden, and A. Muhammad for assistance, especially during COVID-19 times. This study was supported by a grant awarded to S.B. from the German Research Foundation (DFG; BR 3511/4-1). I.K.M. is recipient of a grant from the Stiftungskommission of the LMU Medical Faculty. Research in the S.J. laboratory was supported by Max Planck Society (MPG), Center for Integrated Protein Science Munich (CIPSM), Louis-Jeantet Foundation, and a European Research Council (ERC) Advanced Grant. Research in the S.B., B.P., and A.G.L. laboratories was further supported by the DFG through the collaborative research center CRC 1064 (project ID 213249687-SFB 1064) and by the LMU (A.G.L.) and MPG (B.P.).

## Author contributions

M.C., S.J., and S.B. conceived the study and designed experiments. I.K.M. performed the mammalian experiments. L.M.C. contributed to the microscopy studies in yeast. F.d.B. designed the SUMO F37A I39A mutant. M.C. performed all other experiments. M.C., I.K.M., and L.M.C. analyzed the data. S.B., and S.J. supervised the project. S.B., S.J., B.P., and A.G.L. acquired funding. M.C., and S.B. conceived and wrote the manuscript, and all authors contributed to editing.

## Funding

## Competing interests

The authors declare no competing interests.
