## [Peer Review File · Nature Communications]

Reviewers' Comments:

Reviewer #1:

Remarks to the Author:

Capella et al. investigate the molecular mechanisms by which ribosomal DNA (rDNA) repeats are released from the nucleolus into the nucleoplasm. The authors first observed that disruption of the rDNA tethering complex, composed of the CLIP (Heh1 and Nur1) and the cohibin (Lrs4 and Csm1) complexes, leads to the relocation of rDNA repeats outside of the nucleolus. Using yeast two-hybrids and immunoprecipitation experiments, they identified critical Nur1 domains mediating CLIP-cohibin complex formation. Disruption of the CLIP-cohibin complex releases rDNA from the nucleolus triggering aberrant recombination. They then revealed a multi-step process involving the phosphorylation and SUMOylation of rDNA tethering complex components. Specifically, recognition of SUMOylated Nur1 and Lrs4 proteins by the Cdc48 segregase and its cofactor Ufd1 leads to the rDNA tethering complex's disassembly. Finally, the authors identified that the relocation of damaged rDNA repeats outside of the nucleolus requires the activity of Cdc48/p97 and cofactors in both yeast and human cells.

Overall, Capella et al. identify and report exciting and novel findings that impact our understanding of eukaryotic genome integrity. The answers provided have been long thought after for over a decade. The study was generally well-conducted and presented. Notably, the study provides unique mechanistic insights into the multistep process mediating the relocation and repair of rDNA repeats outside of the nucleolus. With the recommended minor revisions below, the study would be a strong candidate for publication in a leading journal.

1. What is the evidence that the mechanism responsible for rDNA's release in the absence of DNA damage (figs. 2-6) applies to release in its presence (fig. 7)?

2. The rDNA relocation experiments presented in figure 2g are unclear. In the micrograph "outside," it is not clear that the TetI-RFP signal is indeed wholly "outside" of the Nop1-CFP-defined area. One should clearly explain how the microscopy quantifications were performed in figs. 2g, 4c, and 7c. Also, is the rDNA "outside" of the nucleolus associated with DNA damage markers such as Rad52? It is puzzling that the Rad52-YFP was used as a marker in rDNA relocation experiments and is not shown/discussed.

Reviewer #2:

Remarks to the Author:

The ms by Capella et al identifies a role for Cdc48/p97 – a protein that links modified targets to the proteasome, in the repair of damage in the rDNA. As amply demonstrated in a large number of papers, rDNA DSB repair (or fork collapse) takes place after a shift of the damage away from the nucleolus, and to the nuclear pore. The authors seem to have missed this latter very important aspect of the phenomenon and fail to cite relevant literature that puts quite a different spin on their results. While the implication of Cdc48 is essentially their new finding, much of the rest of the story has been shown by others (who are not cited, or not appropriately cited). Moreover, the authors fail to show that the phosphorylation and sumoylation events that modify Lrs4, a protein implicated in rDNA attachment to the INM protein Heh1, actually do trigger a shift in positioning of the rDNA. In two panels, they show a shift "away from the nucleolus" but not the position relative to nuclear pores, in strains bearing Nur1 deletions, or during SUMO overexpression (which causes highly pleiotropic phenotypes and is thus not a particularly revealing condition). However, the authors do not show position relative to the nuclear pore, which multiple studies have implicated in the repair of rDNA lesions. The Cdc48 finding, while interesting, is not sufficient for a Nature Communications paper on this topic.

The weaknesses of this paper could be remedied by complementing their data with studies of recruitment to the nuclear pore, where it has been shown that the repair, desumoylation, etc occurs. The papers (both in yeast and mammalian cells) that they need to cite adequately and integrate into their model, are the following :

Ribosomal RNA gene repeats associate with the nuclear pore complex for maintenance after DNA damage. Horigome C, Unozawa E, Ooki T, Kobayashi T. Horigome C, et al. *PLoS Genet.* 2019 Apr 18;15(4):e1008103. doi: 10.1371/journal.pgen.1008103

Kramarz, K., Schirmeisen, K., Boucherit, V. et al. The nuclear pore primes recombination-dependent DNA synthesis at arrested forks by promoting SUMO removal. *Nat Commun* 11, 5643 (2020). <https://doi.org/10.1038/s41467-020-19516-z>

Liang J, Singh N, Carlson CR, Albuquerque CP, Corbett KD, Zhou H. 2017. Recruitment of a SUMO isopeptidase to rDNA stabilizes silencing complexes by opposing SUMO targeted ubiquitin ligase activity. *Genes Dev* (2017). doi: 10.1101/gad.296145.117

And related studies that are also not cited are:

Relocation of Collapsed Forks to the Nuclear Pore Complex Depends on Sumoylation of DNA Repair Proteins and Permits Rad51 Association. Whalen JM, et al. *Cell Rep.* 2020;31(6):107635. doi: 10.1016/j.celrep.2020.107635.

Pli1(PIAS1) SUMO ligase protected by the nuclear pore-associated SUMO protease Ulp1SEN1/2. Nie M, Boddy MN. Nie M, et al. *J Biol Chem.* 2015 Sep 11;290(37):22678-85. doi: 10.1074/jbc.M115.673038

PolySUMOylation by Siz2 and Mms21 triggers relocation of DNA breaks to nuclear pores through the Slx5/Slx8 STUbL. Horigome C, et al. *Genes Dev.* 2016 30(8):931-45. doi: 10.1101/gad.277665.116

And more recently a bioRxiv preprint first posted online Jan. 24, 2020; doi: <http://dx.doi.org/10.1101/2020.01.24.917203>.

Below is a list going through the paper that points out the missing references to data that would alter their interpretation and help put their findings into context with studies that are leading the field.

In addition, it is quite annoying that the authors use the terms "CLIP" and "Cohibin" when they only monitor one component of a multi- or dimeric-complex. They should refer to Nur1 or Heh1 when they refer to single subunits of the CLIP complex, and the entire paper examines mutants in Lrs4, but not Csm1, yet refers continually to "Cohibin" which is defined as the dimeric complex. Unless they show similar phenotypes for both Csm1 and Lsr4, they should not use the catchphrase Cohibin (which actually has another meaning (i.e. 4-[(Z)-13,14-dihydroxytriacont-17-enyl]-2-methyl-2H-furan-5-one) and therefore is a poor name for the Lsr4 and Csm1). Cohibin is not a universally accepted complex nor is it particularly useful, because each subunit has other activities. Csm1 has multiple activities separate from Lsr4, and Lsr4 actually has activities that cannot be ascribed to Csm1. Given that they do not always work together, it is thus inaccurate and misleading to refer always to Cohibin in this paper, when they manipulate only the Lsr4 gene. That can be easily remedied by revision of the text.

it would be OK to speculate on the extension of the Lsr4 data to Csm1 in the Discussion and there bring up Cohibin.

Detailed points of misinterpretation:

Abstract line 28-29: Note that rDNA damage shifts to nuclear pores, as described in Horigome et al. *PLoS Genet.* 2019, not simply "to the nucleoplasm". One cannot exclude that it goes to the nucleoplasm and then to nuclear pores, but still the site of sumo-targeting is the pore. See also Whalen et al., *Cell Rep.* 2020 – and other refs listed above.

Abstract line 30-31: it is rather absurd to say that the molecular mechanisms of damage relocation are "unknown" when there are a lot of well executed papers and even multiple comprehensive reviews on this very topic. These deal both with yeast and now also mammals (see Kramarz, K et al, *Nat Commun* 2020). Relocation is clearly dependent on sumoylation and on the STUbL Slx5, as

well as on remodelers (papers listed above and earlier work). Recent papers (e.g. Whalen et al) have also addressed which proteins are sumoylated. Thus, this sentence has to be removed or rephrased to state that – “despite a large amount of information on the mechanism of damage relocation, there are still aspects that remain to be explored.” The authors are not well served by ignoring a wealth of data in this field.

Intro line 54: As mentioned above, the use of the term “cohibin” is not very useful. It was originally described as being associated with heterochromatin, but is now shown to bind rDNA specifically (Iglesias et al 2020). Moreover “Cohibin” refers to a chemical compound (Cohibin A and B) widely used. I guess it is acceptable to introduce Cohibin as a heterodimer in the intro, but in the rest of ms, if only Lsr4 is deleted, then the resulting phenotypes should be linked to Lsr4 and not “Cohibin”.

Intro line 74: This is a misstatement: if DSBs in rDNA are kept in the nucleolus, then hyper-recombination with other repeats is promoted leading to deletion or popout of rDNA circles. What they mean is that the loss of NE tethering leads to hyperrecombination. However, the presence of damage in the nucleolus promotes the recombination, the opposite of what is stated (“prevents”). See Torres-Rosell 2007, and doi: 10.1101/gad.296145.117 and doi: 10.1371/journal.pgen.1008103 and refs therein.

Intro line 82: the reloc to pore concerns stalled or collapsed replication forks, DSBs but not simply heterochromatic damage - as the authors imply. Key is the repetitive nature of the damage site. Heterochromatin is already at the nuclear periphery except in flies where centromeric satellite damage shifts to pores – note that *Drosophila* is unique in this. Cite ref 19 and 18, and others more accurately defining these events.

Intro line 85: as mentioned above (and highlighted in the text above) this statement that rDNA relocation is not understood is wrong. Here are four papers that must be cited and summarized at this point. The claims of this current paper must be toned down. Here are papers to cite:

Ribosomal RNA gene repeats associate with the nuclear pore complex for maintenance after DNA damage. Horigome C, Unozawa E, Ooki T, Kobayashi T, Horigome C, et al. PLoS Genet. 2019 Apr 18;15(4):e1008103. doi: 10.1371/journal.pgen.1008103

Kramarz, K., Schirmeisen, K., Boucherit, V. et al. The nuclear pore primes recombination-dependent DNA synthesis at arrested forks by promoting SUMO removal. Nat Commun 11, 5643 (2020). <https://doi.org/10.1038/s41467-020-19516-z>

Liang J, Singh N, Carlson CR, Albuquerque CP, Corbett KD, Zhou H. 2017. Recruitment of a SUMO isopeptidase to rDNA stabilizes silencing complexes by opposing SUMO targeted ubiquitin ligase activity. Genes Dev (2017). doi: 10.1101/gad.296145.117

Torres-Rosell J, et al. Nat Cell Biol. 2007 Aug;9(8):923-31. doi: 10.1038/ncb1619. This paper shows that controlled HR is particularly important for the preservation of repetitive sequences of the rDNA cluster. This paper showed that recombinational repair of a DSB in rDNA in *Saccharomyces cerevisiae* involves the relocalization of the lesion. They also identify SUMOylation players and targets.

And a related paper on the rDNA role of Heh1/Nur1 from *S pombe* is Iglesias N, et al. Native Chromatin Proteomics Reveals a Role for Specific Nucleoporins in Heterochromatin Organization and Maintenance. Mol Cell. 2020;77(1):51-66.e8. doi: 10.1016/j.molcel.2019.10.018

Intro line 91-93: It is not novel to show that sumoylation is involved in the shift of damaged chromatin out of one compartment to another (i.e. to the pore)... I have listed the many papers above that address this. Specifically Torres-Rosell et al, Whalen et al., Horigome et al. 2019; Liang et al., 2017; Horigome et al., 2016. There is also data in flies and mammals (latest is Kramarz et al. 2020). I agree that the fact that “SUMOylated CLIP-cohibin recruits the Cdc48/p97 segregase to assist in 95 the disassembly of the rDNA tethering complex” - is new. However, statements suggesting that this paper is the first to identify sumoylation as the trigger for relocation or even

for Lsr4/Csm1 targeting is misleading. They should show whether the sumoylation targets identified by others are relevant to their proposed pathway, actually.

For the authors' information: extensive information on the sumoylation of the rDNA replication fork barrier complex and its link to Lsr4 and Csm1 is published. Indeed, Csm1 (Srikumar et al. 2013) and its partner, Lrs4, are known to bridge the nuclear envelope to rDNA (Huang et al. 2006) through the Tof2 scaffold protein, which itself is anchored to rDNA by binding to Fob1, which decorates the RFB at each rDNA repeat. This protein network is a sumoylation hot spot and the Ulp2 sumo isopeptidase is responsible for keeping the sumoylation levels of this network in check in the rDNA (Cremona et al. 2012; de Albuquerque et al. 2016; Gillies et al. 2016). Gillies et al. (2016) demonstrated that maintaining balanced sumoylation levels of Tof2, Net1, and Fob1 is important for their association with rDNA. A study by Liang et al. (2017) further addressed how Ulp2 is brought close to its substrates in rDNA. The investigators mapped the Csm1 and Ulp2 interaction regions and determined their structure, showing that Csm1 can interact simultaneously with Ulp2 and Tof2. They solved the structure of the fusion peptides from the three proteins, and performed structure-guided mutagenesis to identify ulp2 mutations that reduce Csm1 binding, as well as a tof2 mutant that reduced Csm1 interaction. Both mutants increased Tof2 sumoylation and lowered rDNA silencing, indicating that Csm1 guides Ulp2 to its target proteins by simultaneous interaction with both the enzyme and target in the rDNA. Furthermore, removing the STUbL or mutating its SIMs suppressed the rDNA silencing defects of ulp2 mutants, by restoring Tof2 levels in ulp2. Read up on this pathway out in : Liang J, et al 2017. Recruitment of a SUMO isopeptidase to rDNA stabilizes silencing complexes by opposing SUMO targeted ubiquitin ligase activity. Genes Dev (2017). doi: 10.1101/gad.296145.117, as it is relevant for their claims about Lsr4 and Csm1 sumoylation.

Results line 174: the statement that loss of tethering increases recombination needs refs. See also doi: 10.1371/journal.pgen.1008103

Results line 190: Here (as elsewhere) the authors jump from a result with Lsr4 to "cohibin" ... rephrase to speak only of Lsr4 unless identical results were obtained with Csm1.

Results line 199 – 201: see papers from Liang et al., and Iglesias et al. 2020.

Results line 219 – 221: There are any number of proteins in the RBP and CLIP complex as well as Lsr4 and Csm1 that are sumoylated as this is (as Liang et al wrote) a "sumoylation hot spot". Their vision needs to include the data from other papers all mentioned above.

Results line 229: Why is it s Sir2-dependent manner and not a Siz2- dependent sumoylation ? Unclear – where was Sir2 deleted in this context ? misstatement ?

Results line 236: From their own data they cannot concluded that "multiple members of the CLIP-cohibin complex are SUMOylated in vivo" but data from many other papers shows this. They must be cited !

Results line 251 -253: This needs to be put into context of other papers looking at the sumoylation of this and related complexes. It is not adequately shown that sumo-Lsr4 compensates for loss of Nur1, as stated. Potentially true but speculative based on current data.

Results line 257 -264: This passage and related data are particularly confusing and the conclusions do not seem justified by the data. First they suggest that sumoylation of Nur1 and Lsr4 are not regulating sister exchange and then argue that if they eliminate SIM recognition they do (?)... then the data showing SUMO overexpression (which is highly pleiotropic) suppresses this is again, not convincing. This passage needs work and conclusions should be toned down.

Results line 267-268: The likely receptors of SUMO at DSBs and in the rDNA at RFP breaks have been elucidated in other papers (which these authors do not appropriately cite) and must be tested or at least discussed in this context. This is a must in a resubmission.

Results line 317 – 322: This section (where breaks are induced) needs to refer to the findings of

Horigome et al 2019, where they show that such damage is shifted to pores.

Refs on sumo link to pores:

Functional targeting of DNA damage to a nuclear pore-associated SUMO-dependent ubiquitin ligase. Nagai S, Dubrana K, Tsai-Pflugfelder M, Davidson MB, Roberts TM, Brown GW, Varela E, Hediger F, Gasser SM, Krogan NJ.. *Science*. 2008 Oct 24;322(5901):597-602. doi: 10.1126/science.1162790.

Nuclear organization in genome stability: SUMO connections.

Nagai S, Davoodi N, Gasser SM. *Cell Res*. 2011 Mar;21(3):474-85. doi: 10.1038/cr.2011.31.

Concerning Heh1/Nur1 : they were earlier implicated in heterochromatin function in *S. pombe* and *S. cerevisiae* (Mekhail et al., 2008; Barrales et al., 2016; Banday et al., 2016). But Iglesias et al., 2020 *Mol Cell* ;77(1):51-66.e8. doi: 10.1016/j.molcel.2019.10.018. Epub 2019 Nov 26) now suggest that these proteins largely interact with the rDNA repeats and heterochromatin-proximal centromeres, rather than heterochromatin per se. The nucleoporins are also associated factors.

In summary, as written, this paper is definitely not publishable due to its lack of integration of the results into the large body of related data across species. A revision should contain the addition of 1) results showing that their "released" damage goes to pores; and 2) testing of several known SIM proteins involved in relocation (Slx5/Slx8) is essential for interpretation of the data. With this an writing that links their results to those of others may render the paper worthy of publication.

Reviewer #3:

Remarks to the Author:

The manuscript entitled "Relocation of rDNA repeats for repair is dependent on SUMO-mediated nucleolar release by the Cdc48/p97 segregase" by Capella and colleagues presents a comprehensive and detailed molecular dissection of the mechanism that controls rDNA stability by regulated release of broken rDNA units to the nucleoplasm for repair by homologous recombination. The manuscript is well written and the data is of high quality and support the conclusions. I have a few issues outlined below that should be addressed before publication and I find it especially important to support more of the conclusions with microscopy data to confirm the inferred effect of various mutants and constructs on rDNA organisation and localization.

Major comments:

1. Fig 1b and 1c: The growth defect resulting from tethering the rDNA to the nuclear periphery is convoluted by Heh1 overexpression by itself causing a growth defect, and GBP-tagging of Heh1 causing an additional growth defect. The authors show that the growth defect caused by HEH1-LRS4 can be reduced by using the weaker GALS promoter. Is it also possible to separating the defects caused by Heh1 overexpression alone and GBP-tagging of Heh1 from the growth defect resulting from artificial tethering observed in Fig 1b by using the GALS promoter?
2. Fig 1b and 1c: How does the rDNA localize upon artificial tethering by the HEH1-LRS4 fusion (using the assay in Fig 2g)?
3. Line 286: The authors equate USCE with release of rDNA and recombination. However, I would expect the opposite. If rDNA is released upon damage then it should be able to undergo equal SCE and not USCE. I think the authors need to test how rDNA breaks localize in mutants that increase or decrease USCE.
4. Figure 5 and line 267-268: Do the various modifications of Nur1 result in the inferred localization of DSBs in the rDNA?

Minor suggestions and corrections:

1. Fig 1b and 1c (or Methods): What concentration of galactose is used?
2. Line 186: It would be helpful with an explanation for the rationale for overexpression of SUMO.
3. Line 162: change "Lsr4" to "Lrs4".
4. In a few places jargon is used instead of more appropriate scientific terms. E.g. "fixed" in line

77. It would be good to eliminate these non-scientific terms.
5. Line 109: change "phenotype" to "defect" to be more precise.
6. Line 398-400: The sentence is unclear. Perhaps it could be explicitly stated that Cdc14 is released from the nucleolus in anaphase.
7. Line 416-419: I find it unlikely that "tethering rDNA repeats permanently to the nuclear envelope is lethal" because of a defect in DSB repair, because yeast cells deficient in both NHEJ and HR show very little loss of fitness in the absence of exogenous genotoxic stress.

GENERAL RESPONSE TO REVIEWER COMMENTS¶

We thank the reviewers for their valuable comments and suggestions for additional experiments and text revisions.

We performed several additional experiments that provided further data to support our original conclusions. We would particularly like to highlight the following new data:

1. **We strengthened our conclusion that recognition of SUMOylation by Cdc48-Ufd1-Npl4 and the resulting CLIP-cohibin dissociation are critical for responding to rDNA damage.** We analyzed rDNA recombination by following *ADE2* marker loss (inserted into rDNA repeats) in mutants lacking the Rrm3 helicase and the replisome component Ctf4, which cause DSBs at rDNA repeats. Combining these mutants with the *nur1KR* mutant (which cannot be SUMOylated) or the *ufd1ΔSIM* mutant (which does not bind SUMO) decreased the *ADE2* loss rate of *rrm3Δ* and *ctf4Δ* single mutants, indicating that rDNA release and recombination is impaired in those double mutants. These results are shown in **Figure 7d**.
2. **New data analysis confirms that Rad52 foci induced by rDNA damage co-localize with extranucleolar rDNA repeats, suggesting that they are repaired by HR in the nucleoplasm.** We followed formation of Rad52 foci and co-localization with rDNA repeats outside the nucleolus following DNA damage. While there are few cells with Rad52 in the absence of DNA damage that rarely co-localize with spontaneous and transient extranucleolar Tet1-mRNA foci, induction of DNA damage results in a marked increase of Rad52 foci that show about 90% co-localization. These results are shown in **Suppl. Figures 7c and 7d**.
3. **New results demonstrate that SUMOylation of Lrs4 triggers rDNA release but is not sufficient to serve as signal for relocation, implying that rDNA release precedes movement to the nuclear periphery.** We analyzed rDNA localization using the Tet1-mRNA marker upon expression of the linear SUMO-Lrs4 fusion by microscopy in- and outside the nucleolus as well as relative to the nuclear periphery using a zoning assay. While we observe that SUMO-Lrs4 expression promotes rDNA release (in addition to CLIP-cohibin dissociation and recombination), we did not observe a relocation of the rDNA marker to the nuclear periphery. These results are shown in **Figure 5f and Suppl. Figure 5f**.
4. **We added data arguing that SUMO overexpression triggers CLIP-cohibin dissociation and rDNA release independently of SUMOylation of Cdc14 (RENT) and Tof2.** Analysis of rDNA association of Cdc14 and Tof2 by ChIP experiments. These results show that Cdc14 and Tof2 abundance at the rDNA repeats is not altered upon SUMO overexpression. Furthermore, we also analyzed steady-state protein levels of Tof2 in cells with and without SUMO overexpression and did not detect any difference. These results are shown in **Suppl. Figures 4e and 4f**.
5. **We further provided data supporting the notion that the rates of rDNA recombination depend on the availability of donor sequences.** Comparison of rDNA recombination using the same assay but in the *uaf30Δ* mutant, which is impaired in RNA Pol I transcription, alleviated the Rad52 exclusion in the nucleolus and increased rDNA recombination inside the nucleolus. As expected, loss of Uaf30 resulted in high recombination rates. Generating double mutants with *nur1Δ* (CLIP) or *csm1Δ* (cohibin) decreased the recombination rate to the level of the *csm1Δ* single mutant, suggesting that rDNA repeat release into the nucleoplasm bypasses recombination inside the nucleolus. The reduced recombination rate in *csm1Δ* and *nur1Δ* confirms that the number of available rDNA repeat donors impacts HR in subnuclear compartments. These results are shown in **Suppl. Figure 3f**.

In addition, we thoroughly revised the text of the manuscript by

- Changing the title to avoid any ambiguity of the scope and focus of our work (new title: *Nucleolar release of rDNA repeats for repair involves SUMO-mediated untethering by the Cdc48/p97 segregase*)
- Restructuring the introduction by adding a new section on relocation of DNA damage sites, their regulation by SUMOylation and rDNA movement to the nuclear periphery.
- Distinguishing between relocation of damaged DNA (to the nuclear periphery) and release of rDNA repeats into the nucleoplasm (from the nucleolus) throughout the paper.
- Adding additional references.

Together, we believe that these changes helped to improve the quality and clarity of our manuscript and thank the reviewers again for their insightful suggestions.

SPECIFIC REPLIES TO INDIVIDUAL REVIEWER COMMENTS¶

Reviewer #1 (Remarks to the Author):

Capella et al. investigate the molecular mechanisms by which ribosomal DNA (rDNA) repeats are released from the nucleolus into the nucleoplasm. The authors first observed that disruption of the rDNA tethering complex, composed of the CLIP (Heh1 and Nur1) and the cohibin (Lrs4 and Csm1) complexes, leads to the relocation of rDNA repeats outside of the nucleolus. Using yeast two-hybrids and immunoprecipitation experiments, they identified critical Nur1 domains mediating CLIP-cohibin complex formation. Disruption of the CLIP-cohibin complex releases rDNA from the nucleolus triggering aberrant recombination. They then revealed a multi-step process involving the phosphorylation and SUMOylation of rDNA tethering complex components. Specifically, recognition of SUMOylated Nur1 and Lrs4 proteins by the Cdc48 segregase and its cofactor Ufd1 leads to the rDNA tethering complex's disassembly. Finally, the authors identified that the relocation of damaged rDNA repeats outside of the nucleolus requires the activity of Cdc48/p97 and cofactors in both yeast and human cells.

Overall, Capella et al. identify and report exciting and novel findings that impact our understanding of eukaryotic genome integrity. The answers provided have been long thought after for over a decade. The study was generally well-conducted and presented. Notably, the study provides unique mechanistic insights into the multistep process mediating the relocation and repair of rDNA repeats outside of the nucleolus. With the recommended minor revisions below, the study would be a strong candidate for publication in a leading journal.

Authors: We thank the reviewer for her/his positive comments.

1. What is the evidence that the mechanism responsible for rDNA's release in the absence of DNA damage (figs. 2-6) applies to release in its presence (fig. 7)?

Authors: Similar to overexpression of SUMO (Fig. 4c), ectopically induced damage following I-Ppol expression causes an increase in formation of rDNA foci outside the nucleolus. This increase depends on the SIM motif of Ufd1 which recognizes SUMOylated proteins (Fig. 7c). In addition, *ADE2* marker loss is increased upon SUMO overexpression (Fig. 4b), whereas *nur1KR* or *ufd1ΔSIM* suppresses marker loss (Supplementary Fig. 5e and Fig. 6g, respectively). To further expand these data, we deleted the helicase *Rrm3* or the chromatin-associated protein *Ctf4*; mutation of these factors is known to induce DSBs at rDNA repeats, similar to I-Ppol (Sasaki and Kobayashi, 2017). As expected, in the absence of *Ctf4* and *Rrm3*, *ADE2* marker loss was strongly increased (new Fig. 7d). Importantly, analogous to what we observed in the in the absence of exogenous DNA damage (Figs. 5e and 6e), we find that increased rDNA recombination is suppressed by the additional mutation *ufd1ΔSIM* or *nur1KR*. Together, this strongly suggests that SUMO-mediated dissociation of the CLIP-cohibin complex resulting in rDNA release (as described in Figs. 2-6) also applies to conditions when additional DNA damage is induced.

2. The rDNA relocation experiments presented in figure 2g are unclear. In the micrograph "outside," it is not clear that the TetI-RFP signal is indeed wholly "outside" of the Nop1-CFP-defined area. One should clearly explain how the microscopy quantifications were performed in figs. 2g, 4c, and 7c. Also, is the rDNA "outside" of the nucleolus associated with DNA damage markers such as Rad52? It is puzzling that the Rad52-YFP was used as a marker in rDNA relocation experiments and is not shown/discussed.

Authors: Thank you for pointing out this lack of clarity about quantification. TetI-mRFP foci either touching or fully overlapping with the Nop1-ECFP nucleolar marker were scored as located at the periphery or inside the nucleolus, respectively, while all other foci were scored as outside the nucleolus. As suggested, we expanded the explanation in the methods section (and refer to the methods in the text). Furthermore, we compared the percentage of cells with Rad52 foci and TetI-mRFP foci outside the nucleolus upon DNA damage and determined the level of co-localization, which is greater than 80% (new Supplementary Fig. 7c). Please note that in the absence of DNA damage, the percentage of cells with Rad52 foci is very low (often less than 20%). Those foci, which likely form independently of transient rDNA relocation, show little correlation with TetI-mRFP localization.

Reviewer #2 (Remarks to the Author):

The ms by Capella et al identifies a role for Cdc48/p97 – a protein that links modified targets to the proteasome, in the repair of damage in the rDNA. As amply demonstrated in a large number of papers, rDNA DSB repair (or fork collapse) takes place after a shift of the damage away from the nucleolus, and to the nuclear pore. The authors seem to have missed this latter very important aspect of the phenomenon and fail to cite relevant literature that puts quite a different spin on their results. While the implication of Cdc48 is essentially their new finding, much of the rest of the story has been shown by others (who are not cited, or not appropriately cited). Moreover, the authors fail to show that the phosphorylation and sumoylation events that modify Lrs4, a protein implicated in rDNA attachment to the INM protein Heh1, actually do trigger a shift in positioning of the rDNA. In two panels, they show a shift “away from the nucleolus” but not the position relative to nuclear pores, in strains bearing Nur1 deletions, or during SUMO overexpression (which causes highly pleiotropic phenotypes and is thus not a particularly revealing condition). However, the authors do not show position relative to the nuclear pore, which multiple studies have implicated in the repair of rDNA lesions. The Cdc48 finding, while interesting, is not sufficient for a Nature Communications paper on this topic.

The weaknesses of this paper could be remedied by complementing their data with studies of recruitment to the nuclear pore, where it has been shown that the repair, desumoylation, etc occurs. The papers (both in yeast and mammalian cells) that they need to cite adequately and integrate into their model, are the following :

Ribosomal RNA gene repeats associate with the nuclear pore complex for maintenance after DNA damage. Horigome C, Unozawa E, Ooki T, Kobayashi T, Horigome C, et al. PLoS Genet. 2019 Apr 18;15(4):e1008103. doi: 10.1371/journal.pgen.1008103

Kramarz, K., Schirmeisen, K., Boucherit, V. et al. The nuclear pore primes recombination-dependent DNA synthesis at arrested forks by promoting SUMO removal. Nat Commun 11, 5643 (2020). <https://doi.org/10.1038/s41467-020-19516-z>

Liang J, Singh N, Carlson CR, Albuquerque CP, Corbett KD, Zhou H. 2017. Recruitment of a SUMO isopeptidase to rDNA stabilizes silencing complexes by opposing SUMO targeted ubiquitin ligase activity. Genes Dev (2017). doi: 10.1101/gad.296145.117

And related studies that are also not cited are:

Relocation of Collapsed Forks to the Nuclear Pore Complex Depends on Sumoylation of DNA Repair Proteins and Permits Rad51 Association. Whalen JM, et al. Cell Rep. 2020;31(6):107635. doi: 10.1016/j.celrep.2020.107635.

Pli1 (PIAS1) SUMO ligase protected by the nuclear pore-associated SUMO protease Ulp1/SEN1/2. Nie M, Boddy MN. Nie M, et al. J Biol Chem. 2015 Sep 11;290(37):22678-85. doi: 10.1074/jbc.M115.673038

PolySUMOylation by Siz2 and Mms21 triggers relocation of DNA breaks to nuclear pores through the Slx5/Slx8 STUbL. Horigome C, et al. Genes Dev. 2016 30(8):931-45. doi: 10.1101/gad.277665.116

And more recently a bioRxiv preprint first posted online Jan. 24, 2020;

doi: <http://dx.doi.org/10.1101/2020.01.24.917203>.

Below is a list going through the paper that points out the missing references to data that would alter their interpretation and help put their findings into context with studies that are leading the field.

Authors: We thank the reviewer for her/his critical comments. We feel that there may have been a conceptual misunderstanding about the molecular mechanism we describe, and we apologize if we were not sufficiently clear and caused confusion by using potentially ambiguous terms.

While rDNA repeats have been described to relocate to the nuclear periphery upon DSB induction (Horigome et al., 2019), they must first be released from the nucleolus, a compartment that precludes repair by HR (Harding et al., 2015; Korsholm et al., 2019; Mooser et al., 2020; van Sluis and McStay, 2015; Torres-Rosell et al., 2007; Warmerdam et al., 2016). Our study aims to unveil the pathway controlling the release from the nucleolus, which has been largely elusive and takes place before relocation to the nuclear periphery. However, we realized that some wording in the previous version of the manuscript may confuse the reader, especially as the term ‘relocation’ has widely been used to describe movement to the nuclear periphery in general and nuclear pore complexes specifically. Hence, we changed the title and the text in order to avoid the term relocation. We instead refer to “nucleolar release” or “movement/release outside the nucleolus”, emphasizing that the mechanism described here acts in steps prior to the relocation to the nuclear periphery. We have also thoroughly revised the introduction to distinguish between movement to the nuclear periphery and nucleolar release. Please note that several of the suggested papers above (Horigome et al., 2019, Ref 23 (revised: Ref 20); Liang et al., 2017, Ref 31 (revised: Ref 34); Horigome et al., 2016, Ref 19 (revised: Ref 12)) were cited in the previous version. We now also included Kramarz et al., 2020 (Ref 16) and Whalen et al., 2020 (Ref 17) and thank you for these suggestions. However, we feel that the report by Nie et al., 2015 addresses findings that are not directly related to our study. We are happy to discuss/explain this last point further if needed.

In addition, it is quite annoying that the authors use the terms “CLIP” and “Cohibin” when they only monitor one component of a multi- or dimeric-complex. They should refer to Nur1 or Heh1 when they refer to single subunits of the CLIP complex, and the entire paper examines mutants in Lrs4, but not Csm1, yet refers continually to “Cohibin” which is defined as the dimeric complex. Unless they show similar phenotypes for both Csm1 and Lrs4, they should not use the catch-phrase Cohibin (which actually has another meaning (i.e. 4-[(Z)-13,14-dihydroxytriacont-17-enyl]-2-methyl-2H-furan-5-one) and therefore is a poor name for the Lrs4 and Csm1). Cohibin is not a universally accepted complex nor is it particularly useful, because each subunit has other activities. Csm1 has multiple activities

separate from Lsr4, and Lsr4 actually has activities that cannot be ascribed to Csm1. Given that they do not always work together, it is thus inaccurate and misleading to refer always to Cohibin in this paper, when they manipulate only the Lsr4 gene. That can be easily remedied by revision of the text. it would be OK to speculate on the extension of the Lsr4 data to Csm1 in the Discussion and there bring up Cohibin.

Authors: As the reviewer suggested, we modified the text and refer only to the particular subunit analyzed in the experiment. We are aware that names of genes, proteins and complexes occasionally have different meanings in another context. However, cohibin and CLIP are frequently used in publications related to rDNA tethering (see PMID: 21664583; PMID: 28487408; PMID: 29959231; PMID: 31553911).

Detailed points of misinterpretation:

Abstract line 28-29: Note that rDNA damage shifts to nuclear pores, as described in Horigome et al. PLoS Genet. 2019, not simply “to the nucleoplasm”. One cannot exclude that it goes to the nucleoplasm and then to nuclear pores, but still the site of sumo-targeting is the pore. See also Whalen et al., Cell Rep. 2020 – and other refs listed above.

Authors: Although damaged rDNA repeats ultimately relocate to the nuclear periphery, they must first be released from the nucleolus into the nucleoplasm, as shown by several studies in yeast and humans (Harding et al., 2015; Korsholm et al., 2019; Mooser et al., 2020; van Sluis and McStay, 2015; Torres-Rosell et al., 2007; Warmerdam et al., 2016). In the revised version, we emphasize that nucleolar release to the nucleoplasm is a first step in repair of broken repeats and likely precedes the following steps, which can happen in the nucleoplasm by Rad52/HR or at the nuclear periphery as part of a “backup system”, as proposed in Horigome et al. (2019).

Abstract line 30-31: it is rather absurd to say that the molecular mechanisms of damage relocation are “unknown” when there are a lot of well executed papers and even multiple comprehensive reviews on this very topic. These deal both with yeast and now also mammals (see Kramarz, K et al, Nat Commun 2020). Relocation is clearly dependent on sumoylation and on the STUbl Slx5, as well as on remodelers (papers listed above and earlier work). Recent papers (e.g. Whalen et al) have also addressed which proteins are sumoylated. Thus, this sentence has to be removed or rephrased to state that – “despite a large amount of information on the mechanism of damage relocation, there are still aspects that remain to be explored.” The authors are not well served by ignoring a wealth of data in this field.

Authors: We apologize for the misunderstanding. We sought to refer here to the release of rDNA repeats from the nucleolus; however, we realize that the choice of words caused misunderstanding. Of course, we fully agree with the reviewer that several studies demonstrated that various types of DNA damage undergo relocation to the nuclear periphery in a SUMO-dependent manner. In the revised version, we rephrase the abstract stating that “Nucleolar release of broken rDNA repeats is conserved from yeast to humans, but the underlying molecular mechanisms are currently unknown” (see lines 30-32), to avoid any confusion. We apologize not being sufficiently clear in the previous version.

Intro line 54: As mentioned above, the use of the term “cohibin” is not very useful. It was originally described as being associated with heterochromatin, but is now shown to bind rDNA specifically (Iglesias et al 2020). Moreover “Cohibin” refers to a chemical compound (Cohibin A and B) widely used. I guess it is acceptable to introduce Cohibin as a heterodimer in the intro, but in the rest of ms, if only Lsr4 is deleted, then the resulting phenotypes should be linked to Lsr4 and not “Cohibin”.

Authors: As suggested, we modified the text and refer exclusively to the particular subunit when describing the observed phenotypes.

Intro line 74: This is a misstatement: if DSBs in rDNA are kept in the nucleolus, then hyper-recombination with other repeats is promoted leading to deletion or popout of rDNA circles. What they mean is that the loss of NE tethering leads to hyperrecombination. However, the presence of damage in the nucleolus promotes the recombination, the opposite of what is stated (“prevents”). See Torres-Rosell 2007, and doi: 10.1101/gad.296145.117 and doi: 10.1371/journal.pgen.1008103 and refs therein.

Authors: The reviewer is correct that presence of a high copy number of repeats would normally drive hyper-recombination. However, in a WT cell, Rad52 recombination foci are excluded from the nucleolus (Torres-Rosell et al., 2007). It has been proposed that deletion of CLIP or cohibin decreases rDNA stability (higher recombination) because “rDNA repeats unleashed from the INM accumulate lesions that can better access the nucleoplasm, in which high concentrations of functional Rad52 promote DNA repair by homologous recombination” (Mekhail et al., 2008). Conversely, deletion of *UAF30* impairs nucleolar exclusion of Rad52 foci, resulting in hyper-recombination (Bernstein et al., 2013). This phenotype is likely due to the presence of high copy numbers of repeats and may be caused by pop-out of rDNA circles, as suggested by the reviewer. We confirmed the hyper-recombination phenotype of *uaf30Δ* cells by testing *ADE2* marker loss. Interestingly, combining *uaf30Δ* with *csm1Δ* or *nur1Δ* did not result in an additional increase but instead decreased the rate of marker loss compared to the *uaf30Δ* single mutant (now shown in Supplementary Fig. 3f). These results indicate the existence of two mutually exclusive repair pathways (in- and outside the nucleolus). The data further supports the notion that loss of CLIP or cohibin promotes the

nucleolar release of broken rDNA repeats into the nucleoplasm. The reduced copy-number of repeats in the nucleoplasm may explain the lower recombination rate of CLIP and cohibin mutant compared with *uaf30Δ*. However, as we become aware that there are likely additional repair mechanisms besides USCE that might trigger marker loss from rDNA repeats (such as SSA), we now refer to recombination or marker loss (instead of USCE) when analyzing rDNA stability.

Intro line 82: the reloc to pore concerns stalled or collapsed replication forks, DSBs but not simply heterochromatic damage - as the authors imply. Key is the repetitive nature of the damage site. Heterochromatin is already at the nuclear periphery except in flies where centromeric satellite damage shifts to pores – note that *Drosophila* is unique in this. Cite ref 19 and 18, and others more accurately defining these events.

Authors: We thank the reviewer for pointing this out. As stated above, we have thoroughly revised the introduction.

Intro line 85: as mentioned above (and highlighted in the text above) this statement that rDNA relocation is not understood is wrong. Here are four papers that must be cited and summarized at this point. The claims of this current paper must be toned down. Here are papers to cite:

Ribosomal RNA gene repeats associate with the nuclear pore complex for maintenance after DNA damage. Horigome C, Unozawa E, Ooki T, Kobayashi T, Horigome C, et al. *PLoS Genet.* 2019 Apr 18;15(4):e1008103. doi: 10.1371/journal.pgen.1008103

Kramarz, K., Schirmeisen, K., Boucherit, V. et al. The nuclear pore primes recombination-dependent DNA synthesis at arrested forks by promoting SUMO removal. *Nat Commun* 11, 5643 (2020). <https://doi.org/10.1038/s41467-020-19516-z>

Liang J, Singh N, Carlson CR, Albuquerque CP, Corbett KD, Zhou H. 2017. Recruitment of a SUMO isopeptidase to rDNA stabilizes silencing complexes by opposing SUMO targeted ubiquitin ligase activity. *Genes Dev* (2017). doi: 10.1101/gad.296145.117

Torres-Rosell J, et al. *Nat Cell Biol.* 2007 Aug;9(8):923-31. doi: 10.1038/ncb1619. This paper shows that controlled HR is particularly important for the preservation of repetitive sequences of the rDNA cluster. This paper showed that recombinational repair of a DSB in rDNA in *Saccharomyces cerevisiae* involves the relocalization of the lesion. They also identify SUMOylation players and targets.

And a related paper on the rDNA role of Heh1/Nur1 from *S pombe* is Iglesias N, et al. Native Chromatin Proteomics Reveals a Role for Specific Nucleoporins in Heterochromatin Organization and Maintenance. *Mol Cell.* 2020;77(1):51-66.e8. doi: 10.1016/j.molcel.2019.10.018

Authors: The reviewer is correct and much is known about relocation of DSBs to the nuclear periphery and NPCs. However, to our knowledge, the molecular mechanism triggering nucleolar release of rDNA repeats is largely unknown. We rephrased the statement to avoid confusion: “However the regulatory mechanisms controlling the initial release of broken rDNA repeats from the nucleolus remain largely elusive”. Please note that we cited some of the suggested publications in the introduction of the original manuscript (e.g., in lines 69, 71, 84). We now also cite Kramarz et al. 2020, as suggested.

Intro line 91-93: It is not novel to show that sumoylation is involved in the shift of damaged chromatin out of one compartment to another (i.e. to the pore)... I have listed the many papers above that address this. Specifically Torres-Rosell et al, Whalen et al., Horigome et al. 2019; Liang et al., 2017; Horigome et al., 2016. There is also data in flies and mammals (latest is Kramarz et al. 2020). I agree that the fact that “SUMOylated CLIP-cohibin recruits the Cdc48/p97 segregase to assist in the disassembly of the rDNA tethering complex” - is new. However, statements suggesting that this paper is the first to identify sumoylation as the trigger for relocation or even for Lsr4/Csm1 targeting is misleading. They should show whether the sumoylation targets identified by others are relevant to their proposed pathway, actually.

Authors: We fully agree that SUMO is important for chromatin movement from one compartment to another upon DNA damage. However, here we describe the molecular mechanisms occurring within the nucleolus and that precede relocation to the nuclear periphery. These events include phosphorylation and SUMOylation, which trigger step-wise disassembly of the rDNA tethering complex and the movement of the rDNA repeats outside the nucleolus. Please note that although damaged rDNA repeats have been shown to be relocated to the nuclear periphery (Horigome et al., 2019), a role for SUMO in this process was not addressed by the authors. Here we link two SUMO-modified factors to nucleolar release, a process that likely takes place before relocation to the nuclear envelope. We find that SUMOylation of Lrs4 increases the frequency of marked repeats outside the nucleolus (new Fig. 5f) and rDNA recombination (new Fig. 5e), without affecting the enrichment of rDNA repeats at the nuclear periphery (new Supplementary Fig. 5f). Additionally, we show that blocking Nur1 SUMOylation decreases rDNA recombination (new Figs. 5d and 7d, and Supplementary Fig. 5e), and impairs rDNA movement outside the nucleolus upon DNA damage (new Fig. 7c).

Most of the described SUMOylated factors involved in chromatin movement from one compartment to another are required for the relocation of persistent DSBs to the nuclear periphery. However, since our proposed mechanism likely happens prior the movement to nuclear pores, we feel that analyzing these SUMOylation targets in rDNA release is beyond the scope of this manuscript. Nonetheless, in the revised version, we refrain from using the term “relocation” but instead refer to “nucleolar release” to avoid any potential misconception about the molecular process investigated in our study.

For the authors' information: extensive information on the sumoylation of the rDNA replication fork barrier complex and its link to Lsr4 and Csm1 is published. Indeed, Csm1 (Srikumar et al. 2013) and its partner, Lrs4, are known to bridge the nuclear envelope to rDNA (Huang et al. 2006) through the Tof2 scaffold protein, which itself is anchored to rDNA by binding to Fob1, which decorates the RFB at each rDNA repeat. This protein network is a sumoylation hot spot and the Ulp2 sumo isopeptidase is responsible for keeping the sumoylation levels of this network in check in the rDNA (Cremona et al. 2012; de Albuquerque et al. 2016; Gillies et al. 2016). Gillies et al. (2016) demonstrated that maintaining balanced sumoylation levels of Tof2, Net1, and Fob1 is important for their association with rDNA. A study by Liang et al. (2017) further addressed how Ulp2 is brought close to its substrates in rDNA. The investigators mapped the Csm1 and Ulp2 interaction regions and determined their structure, showing that Csm1 can interact simultaneously with Ulp2 and Tof2. They solved the structure of the fusion peptides from the three proteins, and performed structure-guided mutagenesis to identify *ulp2* mutations that reduce Csm1 binding, as well as a *tof2* mutant that reduced Csm1 interaction. Both mutants increased Tof2 sumoylation and lowered rDNA silencing, indicating that Csm1 guides Ulp2 to its target proteins by simultaneous interaction with both the enzyme and target in the rDNA. Furthermore, removing the STUbL or mutating its SIMs suppressed the rDNA silencing defects of *ulp2* mutants, by restoring Tof2 levels in *ulp2*. Read up on this pathway out in : Liang J, et al 2017. Recruitment of a SUMO isopeptidase to rDNA stabilizes silencing complexes by opposing SUMO targeted ubiquitin ligase activity. *Genes Dev* (2017). doi: 10.1101/gad.296145.117, as it is relevant for their claims about Lsr4 and Csm1 sumoylation.

Authors: We thank the reviewer for pointing out the role of the SUMO isopeptidase Ulp2 in controlling SUMOylation of various factors associated with rDNA repeats including Tof2. We are aware of the study by Liang et al. (2017), where the authors showed that failure to recruit Ulp2 to the nucleolus increases SUMOylation and degradation of Tof2, which in return impairs rDNA silencing. In agreement with this study, we already showed in our previous version (REF figure) that expressing a C-terminally truncated mutant of Ulp2 (*ulp2ΔC*) deficient in nucleolar localization results in increased SUMOylation of nucleolar factors and display increased rDNA marker loss. While we cannot exclude that increased recombination in *ulp2ΔC* cells is caused by deficient rDNA silencing, we found that SUMO overexpression also increases rDNA recombination but without affecting rDNA silencing (analyzed for NTS1 and NTS2 expression; previously Supplementary Fig. 4c; new Supplementary Fig. 4d). In the new version, we further show that overexpression of SUMO does not affect Tof2 protein stability and/or its binding to rDNA repeats (new Supplementary Fig. 4e,f), known to be affected in *ulp2* mutants (Gillies et al., 2016; Liang et al., 2017). Moreover, while Liang et al. (2017) showed that Csm1 recruits Ulp2, they did not detect SUMOylated Lrs4 or Csm1 in *ulp2* mutant cells, suggesting that both might be only modified upon specific conditions, such as rDNA damage. Thus, we believe that the SUMO-dependent mechanism we describe for rDNA release is different from the one described by Liang et al. (2017).

Results line 174: the statement that loss of tethering increases recombination needs refs. See also doi: 10.1371/journal.pgen.1008103

Authors: We thank the reviewer for pointing this out. We have included the suggested reference.

Results line 190: Here (as elsewhere) the authors jump from a result with Lsr4 to “cohibin” ... rephrase to speak only of Lsr4 unless identical results were obtained with Csm1.

Authors: Thank you. We corrected this.

Results line 199 – 201: see papers from Liang et al., and Iglesias et al. 2020.

Authors: We thank the reviewer for this suggestion. In the previous version, we showed that overexpression of SUMO increases rDNA instability yet without affecting silencing (previously Supplementary Fig. 4c, now Supplementary Fig. 4d), as shown in Liang et al. (2017) for *ulp2ΔC* cells. In the revised version, we include additional data related to other rDNA factors modified by SUMO (see response below).

Results line 219 – 221: There are any number of proteins in the RBP and CLIP complex as well as Lsr4 and Csm1 that are sumoylated as this is (as Liang et al wrote) a “sumoylation hot spot”. Their vision needs to include the data from other papers all mentioned above.

Authors: The reviewer is correct that several rDNA factors have been shown to be SUMOylated (e.g., Fob1, Tof2 and members of the RENT complex) and that deleting *ULP2* results in increased SUMOylation of these factors, which impairs their binding to rDNA repeats or degradation of Tof2 (Gillies et al., 2016; Liang et al., 2017). We agree that it is important to rule out potential indirect effects. In the revised version, we demonstrate that the enrichment of Tof2 or Cdc14 (RENT subunit) at rDNA is unaltered in cells overexpressing SUMO (new Supplementary Fig. 4e). Moreover, Tof2 protein levels remain stable when SUMO is overexpressed (new Supplementary Fig. 4f). Based on these results, we conclude that SUMO-mediated rDNA release from the nucleolus is independent of SUMOylation of Tof2 and RENT.

Results line 229: Why is it s Sir2-dependent manner and not a Siz2- dependent sumoylation ? Unclear – where was Sir2 deleted in this context ? misstatement ?

Authors: In the previous version, we showed that Nur1 SUMOylation depends on the presence of Sir2 (previously Supplementary Fig. 5b,c; new Supplementary Fig. 5a,b). However, we stated that Nur1 modification requires the activity of the E3 ligase Siz2 (shown in Fig. 5a) and depends on the association of CLIP to the rDNA (shown in Fig. 5b and Supplementary Fig. 5a,b; and related to Mekhail et al., 2008).

Results line 236: From their own data they cannot concluded that “multiple members of the CLIP-cohibin complex are SUMOylated in vivo” but data from many other papers shows this. They must be cited !

Authors: As suggested, we included the corresponding references.

Results line 251 -253: This needs to be put into context of other papers looking at the sumoylation of this and related complexes. It is not adequately shown that sumo-Lsr4 compensates for loss of Nur1, as stated. Potentially true but speculative based on current data.

Authors: As suggested, we toned down our conclusions.

Results line 257 -264: This passage and related data are particularly confusing and the conclusions do not seem justified by the data. First they suggest that sumoylation of Nur1 and Lsr4 are not regulating sister exchange and then argue that if they eliminate SIM recognition they do (?)... then the data showing SUMO overexpression (which is highly pleiotropic) suppresses this is again, not convincing. This passage needs work and conclusions should be toned down.

Authors: In agreement with previous reports (Chung et al., 2015; Corbett et al., 2010; Godfrey et al., 2015), we show that deletion of *NUR1* and *LRS4* increases rDNA recombination using the *ADE2* marker inserted into rDNA repeats. To examine the presence of SUMO in the CLIP-cohibin complex, we introduced SUMO fusions of Nur1 or Lrs4 into the corresponding deletion mutant (Fig. 5e). Unlike the non-fused WT versions of Nur1 or Lrs4, the fusion proteins failed to complement the deletion mutant phenotype, resulting in increased marker loss similar to the empty vector. Hence, we conclude that permanent presence of SUMO at Nur1 or Lrs4 disrupts the CLIP-cohibin complex. However, increased recombination is not seen under this condition when the SUMO interaction motif (SIM) of Ufd1 is absent (Fig. 6g,h) or when linear fusions with a mutant SUMO moiety that cannot be bound by Ufd1 is expressed (Fig. 5f). We find a similar functional relationship when we overexpress SUMO, which produces a similar phenotype as seen for the linear SUMO fusion proteins. Please note that while SUMO overexpression may indeed cause pleiotropic effects, increased *ADE2* marker loss under overexpression conditions strictly depends on the presence of the SIM in Ufd1 (Fig. 6g, right panel), arguing that recombination of rDNA repeats is related to ability of Cdc48-Ufd1-Npl4 to recognize SUMOylated proteins. We have revised the corresponding section in the text to reduce potential confusion.

Results line 267-268: The likely receptors of SUMO at DSBs and in the rDNA at RFP breaks have been elucidated in other papers (which these authors do not appropriately cite) and must be tested or at least discussed in this context. This is a must in a resubmission.

Authors: Indeed, additional factors implicated in the relocation of persistent DNA damage to the nuclear periphery have been shown to be SUMOylated. We tested whether expression of linear SUMO-Lrs4 results in increased frequency of rDNA repeats at the nuclear envelope using Nup49 as a NE marker (new Supplementary Fig. 5f). Interestingly, while we see that this linear fusion causes increased recombination (Fig. 5e), we don't observe a significant change in NE localization, suggesting that SUMOylation of Lrs4 is relevant for CLIP-cohibin disruption and rDNA release from the nucleolus, but not its relocation to the NE. We presume that other SUMO targets are involved in the NE relocation step. We feel that testing those additional SUMO-receptors at DSBs is beyond the scope of the present study, since we exclusively focus on nucleolar rDNA release. Nevertheless, we now mention that SUMOylation of several factors is required for the DSBs relocation (see lines 70-80 and 297-299), as suggested by the reviewer.

Results line 317 – 322: This section (where breaks are induced) needs to refer to the findings of Horigome et al 2019, where they show that such damage is shifted to pores.

Authors: We now refer to the findings by Horigome et al. (2019) discussing whether SUMOylation of Lrs4 might trigger/facilitate the relocation of rDNA repeats to the nuclear periphery (lines 297-299). Please note that Lrs4 modification does not affect rDNA movement to the nuclear periphery (new Supplementary Fig. 5f, described above). In the section mentioned by the reviewer (previous lines 317-322), we examined the release of broken repeats out of the nucleolus, which is independent and upstream of relocation to the nuclear periphery and seems unrelated to the process the reviewer refers to. Hence, referring to this publication does not seem necessary.

Refs on sumo link to pores:

Functional targeting of DNA damage to a nuclear pore-associated SUMO-dependent ubiquitin ligase. Nagai S, Dubrana K, Tsai-Pflugfelder M, Davidson MB, Roberts TM, Brown GW, Varela E, Hediger F, Gasser SM, Krogan NJ. *Science*. 2008 Oct 24;322(5901):597-602. doi: 10.1126/science.1162790.

Nuclear organization in genome stability: SUMO connections.

Nagai S, Davoodi N, Gasser SM. *Cell Res*. 2011 Mar;21(3):474-85. doi: 10.1038/cr.2011.31.

Concerning Heh1/Nur1 : they were earlier implicated in heterochromatin function in *S. pombe* and *S. cerevisiae* (Mekhail et al., 2008; Barrales et al., 2016; Banday et al., 2016). But Iglesias et al., 2020 *Mol Cell* ;77(1):51-66.e8. doi: 10.1016/j.molcel.2019.10.018. Epub 2019 Nov 26) now suggest that these proteins largely interact with the rDNA repeats and heterochromatin-proximal centromeres, rather than heterochromatin per se. The nucleoporins are also associated factors.

In summary, as written, this paper is definitely not publishable due to its lack of integration of the results into the large body of related data across species. A revision should contain the addition of 1) results showing that their "released" damage goes to pores; and 2) testing of several known SIM proteins involved in relocation (Slx5/Slx8) is essential for interpretation of the data. With this an writing that links their results to those of others may render the paper worthy of publication.

Authors: As stated above, the aim of this work is to understand the molecular mechanisms triggering the nucleolar release of broken rDNA repeats, a process conserved from yeast to humans. Although damaged repeats are relocated to the nuclear periphery (Horigome et al., 2019), we propose that the molecular events revealed by our study are acting before the movement to the nuclear periphery takes place. In agreement, expression of SUMO-Lrs4 increased rDNA recombination and nucleolar release, but did not affect the frequency of marked rDNA repeats at the nuclear periphery (new Fig. 5e,f and Supplementary Fig. 5f).

Regarding the role of STUbLs, we showed in the previous version that deletion of *SLX5/SLX8* or *ULS1* does not affect the interaction between Nur1 and Lrs4 (previously Supplementary Fig. 6a,b; new Supplementary Fig. 6b,c), suggesting that the STUbLs do not regulate the nucleolar release of rDNA repeats. However, we cannot completely exclude that the STUbLs might play a role in a later stage. Previously, we showed that the STUbL RNF4 is required for rDNA segregation in humans (previous Fig. 7e, new Fig. 8b), and acts in the same pathway as p97-NPLOC4-UFD1L. It is noteworthy that, unlike in yeast, the human p97-NPLOC4-UFD1L complex has not been shown to interact with SUMO. In the new version, we show that overexpression of sumoKRall (which cannot form SUMO/ubiquitin chains) in yeast cells fosters the interaction between Nur1 and Lrs4, and decreases rDNA recombination (new Fig. 5g and Supplementary Fig. 6a). Together, these data suggest that modification of SUMO, either by polyubiquitination or polySUMOylation, is required for nucleolar release of rDNA repeats.

Again, we apologize if we failed to appropriately describe the underlying processes or used ambiguous terminology. We thoroughly revised the text and hope that the revised version and the additional series of experiments will convince this reviewer of the novelty of our findings.

Reviewer #3 (Remarks to the Author):

The manuscript entitled "Relocation of rDNA repeats for repair is dependent on SUMO-mediated nucleolar release by the Cdc48/p97 segregase" by Capella and colleagues presents a comprehensive and detailed molecular dissection of the mechanism that controls rDNA stability by regulated release of broken rDNA units to the nucleoplasm for repair by homologous recombination. The manuscript is well written and the data is of high quality and support the conclusions. I have a few issues outlined below that should be addressed before publication and I find it especially important to support more of the conclusions with microscopy data to confirm the inferred effect of various mutants and constructs on rDNA organisation and localization.

Authors: We thank the reviewer for her/his positive comments.

Major comments:

1. Fig 1b and 1c: The growth defect resulting from tethering the rDNA to the nuclear periphery is convoluted by Heh1 overexpression by itself causing a growth defect, and GBP-tagging of Heh1 causing an additional growth defect. The authors show that the growth defect caused by HEH1-LRS4 can be reduced by using the weaker GALS promoter. Is it also possible to separating the defects caused by Heh1 overexpression alone and GBP-tagging of Heh1 from the growth defect resulting from artificial tethering observed in Fig 1b by using the GALS promoter?

Authors: Thank you for pointing this out. Indeed, Heh1 overexpression itself causes a growth defect. For better comparison and as suggested, we show *pGAL1-HEH1* (or *pGALS-HEH1*) as reference strain rather than the strain harboring the empty plasmid. We have changed the corresponding figures (new Fig. 1b and Supplementary Fig. 1c).

2. Fig 1b and 1c: How does the rDNA localize upon artificial tethering by the HEH1-LRS4 fusion (using the assay in Fig 2g)?

Authors: As suggested, we examined rDNA location relative to the nucleolus in cells constantly expressing Heh1, Heh1-Lrs4 or Heh1-Lrs4 Q325Stop with the *GALS* promoter. While we did not find a higher frequency of cells with rDNA foci *within* the nucleolus in Heh1-Lrs4 compared to Heh1 strains, we observed increased rDNA release upon expression of Heh1-Lrs4 Q325Stop. When interpreting these results, it should be noted that nucleolar localization of the rDNA marker is already high in cells expressing non-fused Heh1 (i.e., more than 85% foci are inside). Thus, a further increase in nuclear localization under conditions of artificial tethering might be difficult to detect because wild-type Heh1 is still expressed. The results are shown in the new Supplementary Fig. 1f.

3. Line 286: The authors equate USCE with release of rDNA and recombination. However, I would expect the opposite. If rDNA is released upon damage then it should be able to undergo equal SCE and not USCE. I think the authors need to test how rDNA breaks localize in mutants that increase or decrease USCE.

Authors: While marker loss of *ADE2* at rDNA repeats was referred to as USCE in a previous study (Mekhail et al., 2008), the reviewer is correct: USCE for rDNA repair is most likely to occur when repeats are kept inside the nucleolus. To discriminate between possible repair mechanisms inside and outside of the nucleolus, we took advantage of a mutant, *uaf30Δ*, which impairs nucleolar exclusion of Rad52 recombination foci and promotes repair inside the nucleolus, thus bypassing the need for nucleolar release (Bernstein et al., 2013). In agreement with previous studies, we found high rates of *ADE2* marker loss in the *uaf30Δ* mutant. Interestingly, when we combine this mutant with deletions of *CSM1* or *NUR1*, which themselves cause hyper-recombination (although not to the high levels as seen in *uaf30Δ*), instead of an additional increase we observe a decrease to similar levels as seen in the single *csm1Δ* mutant (new Supplementary Fig. 3e). These results suggest the existence of two mutually exclusive repair pathways (in- and outside the nucleolus), further supporting the notion that loss of CLIP or cohibin promotes the nuclear release of broken rDNA repeats into the nucleoplasm. The reduced copy-number of repeats in the nucleoplasm may explain the lower recombination rate observed in the CLIP and cohibin mutant compared to the *uaf30Δ* strain. However, being aware of possible additional repair mechanisms besides USCE that might trigger marker loss from rDNA repeats (such as SSA), we now refer to this process as recombination or marker loss (instead of USCE) when analyzing rDNA stability.

4. Figure 5 and line 267-268: Do the various modifications of Nur1 result in the inferred localization of DSBs in the rDNA?

Authors: This is indeed an interesting question. We showed before that, under normal growth conditions, removal of the Nur1 C-terminus, which contains relevant phosphorylation sites, increases rDNA release to the nucleoplasm (Fig. 2f), but does not alter release upon damage (Fig. 7c). A caveat to this approach is that the C-terminus of Nur1 is also required for its interaction with cohibin. Hence, we cannot make a specific statement about the role of phosphorylation with this assay. However, we now analyze the *nur1KR* mutant and find that it partially blocks rDNA release from the nucleolus upon DNA damage. Since Nur1 is likely not the only target of the CLIP-cohibin complex that is modified by SUMO, this may explain why the block is only partial. The results are shown in the new Fig. 7c and Supplementary Fig. 7c.

Minor suggestions and corrections:

1. Fig 1b and 1c (or Methods): What concentration of galactose is used?

Authors: We used galactose 2% (final concentration). This information is now included in the corresponding figure legends (see lines 936 and 1096).

2. Line 186: It would be helpful with an explanation for the rationale for overexpression of SUMO.

Authors: Under normal growth conditions it is likely that only a small proportion of a given target is SUMO-modified. This can be overcome by SUMO overexpression, which increases the amount of SUMO conjugates (see Supplementary Fig. S4a). As the reviewer suggested, we included the following statement in the results: "Since SUMO targets may be not or only partially modified in the absence of DNA damage, we decided to overexpress SUMO to increase the SUMOylation steady-state levels (Supplementary Fig. 4a)" (see lines 218-221).

3. Line 162: change "Lsr4" to "Lrs4".

Authors: Thank you for spotting the typo. Corrected.

4. In a few places jargon is used instead of more appropriate scientific terms. E.g. "fixed" in line 77. It would be good to eliminate these non-scientific terms.

Authors: Thank you for pointing this out. Throughout the revised version, we changed the text to avoid any jargon.

5. Line 109: change "phenotype" to "defect" to be more precise.

Authors: Done.

6. Line 398-400: The sentence is unclear. Perhaps it could be explicitly stated that Cdc14 is released from the nucleolus in anaphase.

Authors: As suggested, we rephrased the sentence in order to make it clearer: "Nur is phosphorylated in a cell cycle-dependent manner by the cyclin-dependent kinase Cdk1, which is counteracted by the phosphatase Cdc14. Interestingly, phosphorylated Nur1 inhibits Cdc14 activation, and promotes Cdc14 release from the nucleolus once dephosphorylated in early anaphase" (see lines 453-456).

7. Line 416-419: I find it unlikely that "tethering rDNA repeats permanently to the nuclear envelope is lethal" because of a defect in DSB repair, because yeast cells deficient in both NHEJ and HR show very little loss of fitness in the absence of exogenous genotoxic stress.

Authors: We thank the reviewer for her/his constructive comment. We changed the text accordingly (see lines 471-482).

References:

- Bernstein, K.A., Juanchich, A., Sunjevaric, I., and Rothstein, R. (2013). The Shu complex regulates Rad52 localization during rDNA repair. *DNA Repair (Amst)*. *12*, 786–790.
- Chung, D.K.C., Chan, J.N.Y., Strecker, J., Zhang, W., Ebrahimi-Ardebili, S., Lu, T., Abraham, K.J., Durocher, D., and Mekhail, K. (2015). Perinuclear tethers license telomeric DSBs for a broad kinesin- and NPC-dependent DNA repair process. *Nat. Commun.* *6*.
- Corbett, K.D., Yip, C.K., Ee, L.S., Walz, T., Amon, A., and Harrison, S.C. (2010). The monopolin complex crosslinks kinetochore components to regulate chromosome-microtubule attachments. *Cell* *142*, 556–567.
- Gillies, J., Hickey, C.M., Su, D., Wu, Z., Peng, J., and Hochstrasser, M. (2016). SUMO pathway modulation of regulatory protein binding at the ribosomal dna locus in *Saccharomyces cerevisiae*. *Genetics* *202*, 1377–1394.
- Godfrey, M., Kuilman, T., and Uhlmann, F. (2015). Nur1 Dephosphorylation Confers Positive Feedback to Mitotic Exit Phosphatase Activation in Budding Yeast. *PLoS Genet.* *11*.
- Harding, S.M., Boiarsky, J.A., and Greenberg, R.A. (2015). ATM Dependent Silencing Links Nucleolar Chromatin Reorganization to DNA Damage Recognition. *Cell Rep.* *13*, 251–259.
- Horigome, C., Unozawa, E., Ooki, T., and Kobayashi, T. (2019). Ribosomal RNA gene repeats associate with the nuclear pore complex for maintenance after DNA damage. *PLOS Genet.* *15*, e1008103.
- Korsholm, L.M., Gál, Z., Lin, L., Quevedo, O., Ahmad, D.A., Dulina, E., Luo, Y., Bartek, J., and Larsen, D.H. (2019). Double-strand breaks in ribosomal RNA genes activate a distinct signaling and chromatin response to facilitate nucleolar restructuring and repair. *Nucleic Acids Res.* *47*, 8019–8035.
- Liang, J., Singh, N., Carlson, C.R., Albuquerque, C.P., Corbett, K.D., and Zhou, H. (2017). Recruitment of a SUMO isopeptidase to rDNA stabilizes silencing complexes by opposing SUMO targeted ubiquitin ligase activity. *Genes Dev.* *31*, 802–815.
- Lisby, M., Rothstein, R., and Mortensen, U.H. (2001). Rad52 forms DNA repair and recombination centers during S phase. *Proc. Natl. Acad. Sci. U. S. A.* *98*, 8276–8282.
- Mekhail, K., Seebacher, J., Gygi, S.P., and Moazed, D. (2008). Role for perinuclear chromosome tethering in maintenance of genome stability. *Nature* *456*, 667–670.
- Mooser, C., Symeonidou, I.E., Leimbacher, P.A., Ribeiro, A., Shorrock, A.M.K., Jungmichel, S., Larsen, S.C., Knechtle, K., Jasrotia, A., Zurbriggen, D., et al. (2020). Treacle controls the nucleolar response to rDNA breaks via TOPBP1 recruitment and ATR activation. *Nat. Commun.* *11*, 1–16.
- Sasaki, M., and Kobayashi, T. (2017). Ctf4 Prevents Genome Rearrangements by Suppressing DNA Double-Strand Break Formation and Its End Resection at Arrested Replication Forks. *Mol. Cell* *66*, 533-545.e5.
- van Sluis, M., and McStay, B. (2015). A localized nucleolar DNA damage response facilitates recruitment of the homology-directed repair machinery independent of cell cycle stage. *Genes Dev.* *29*, 1151–1163.
- Torres-Rosell, J., Sunjevaric, I., De Piccoli, G., Sacher, M., Eckert-Boulet, N., Reid, R., Jentsch, S., Rothstein, R., Aragón, L., and Lisby, M. (2007). The Smc5-Smc6 complex and SUMO modification of Rad52 regulates recombinational repair at the ribosomal gene locus. *Nat. Cell Biol.* *9*, 923–931.
- Warmerdam, D.O., van den Berg, J., and Medema, R.H. (2016). Breaks in the 45S rDNA Lead to Recombination-Mediated Loss of Repeats. *Cell Rep.* *14*, 2519–2527.

Reviewers' Comments:

Reviewer #1:

Remarks to the Author:

The authors have satisfactorily addressed my original comments, the manuscript is further improved, and I support publication. I have one minor comment that the authors should consider while finalizing the paper.

The authors have indicated that the ADE2 marker loss assay can reveal different types of recombinations. While that is correct, as used in this study to quantify the fraction of half-sectored colonies, the assay is measuring USCE events occurring at the first cell division after plating the cells. Other types of recombination events at that time point would not yield half-sectored colonies. However, the use of this assay to count any partially red colonies can indeed reveal other types of marker loss and recombinations, but this does not appear to be what was done here, correct? So it is important to be precise and reach an interpretation reflecting the exact quantification approach.

Reviewer #3:

Remarks to the Author:

The revised manuscript entitled "Nucleolar release of rDNA repeats for repair involves SUMO-mediated untethering by the Cdc48/p97 segregase" by Capella and colleagues have addressed my concerns and questions for the original manuscript. However, I have identified a few minor corrections that I suggest to implement before publication:

1. Line 308: Delete "of".
2. Line 146: I do not find it convincingly established that cell viability is causally related to "the dissociation of ribosomal repeats from the nuclear periphery and nucleolus, and dynamic CLIP-cohibin disassembly". The events that lead to cell lethality or slow growth in figure 1 are not identified in the paper. I therefore suggest to change the wording "critically depends on" to "correlates with".
3. Line 363: DSBs are not created in every rDNA repeat by I-PpoI, so I suggest to change "which generates DSBs" to "which potentially generates DSBs".
4. Line 453: change "Nur" to "Nur1".

SPECIFIC REPLIES TO INDIVIDUAL REVIEWER COMMENTS¶

Reviewer #1 (Remarks to the Author):

The authors have satisfactorily addressed my original comments, the manuscript is further improved, and I support publication. I have one minor comment that the authors should consider while finalizing the paper.

Authors: We thank the reviewer for her/his positive comments.

The authors have indicated that the ADE2 marker loss assay can reveal different types of recombinations. While that is correct, as used in this study to quantify the fraction of half-sectored colonies, the assay is measuring USCE events occurring at the first cell division after plating the cells. Other types of recombination events at that time point would not yield half-sectored colonies. However, the use of this assay to count any partially red colonies can indeed reveal other types of marker loss and recombinations, but this does not appear to be what was done here, correct? So it is important to be precise and reach an interpretation reflecting the exact quantification approach.

Authors: We appreciate the constructive comment of the reviewer. The rates of marker loss from rDNA repeats have been referred to as both USCE (Abraham et al., 2019 PMID: 31776241; Mekhail et al., 2008 PMID: 18997772; Salvi et al., 2014 PMID: 25073155) or rDNA recombination (Bernstein et al., 2013 PMID: 23790361; Ha et al., 2014 PMID: 24981510; Kaeberlein et al., 1999 PMID: 10521401; Menzel et al., 2014 PMID: 24361936; Torres-Rosell et al., 2007 PMID: 17643116). Several repair mechanisms by recombination could lead to marker loss. However, since only half-sectored colonies were quantified to assess rDNA recombination, the revised version states that rDNA marker loss through USCE is quantified, both in the main text and the methods section.

Reviewer #3 (Remarks to the Author):

The revised manuscript entitled “Nucleolar release of rDNA repeats for repair involves SUMO-mediated untethering by the Cdc48/p97 segregase“ by Capella and colleagues have addressed my concerns and questions for the original manuscript.

Authors: We thank the reviewer for her/his positive comments.

However, I have identified a few minor corrections that I suggest to implement before publication:

1. Line 308: Delete “of”.

Authors: Done.

2. Line 146: I do not find it convincingly established that cell viability is causally related to “the dissociation of ribosomal repeats from the nuclear periphery and nucleolus, and dynamic CLIP-cohibin disassembly”. The events that lead to cell lethality or slow growth in figure 1 are not identified in the paper. I therefore suggest to change the wording “critically depends on” to “correlates with”.

Authors: The reviewer is correct, and the causes of the slow growth generated by the permanent tethering were not fully identified. As suggested, we replaced “critically depends on” by “correlates with”.

3. Line 363: DSBs are not created in every rDNA repeat by I-PpoI, so I suggest to change “which generates DSBs” to “which potentially generates DSBs”.

Authors: We thank the reviewer for her/his constructive comment. We changed the text accordingly.

4. Line 453: change “Nur” to “Nur1”.

Authors: Thank you for spotting the typo. Corrected.